

**Assessing Carbon Dioxide Removal Through Global and Regional Ocean Alkalization**
**under High and Low Emission Pathways.**
Andrew Lenton [1,2], Richard J. Matear [1], David P. Keller [3], Vivian Scott[4] and Naomi E.
Vaughan[5]
[1] CSIRO Oceans and Atmosphere, Hobart Australia
[2] Antarctic Climate and Ecosystems Co-operative Research Centre, Hobart, Australia
[3] GEOMAR Helmholtz Centre for Ocean Research Kiel, Germany
[4] School of Geosciences, University of Edinburgh, Edinburgh, United Kingdom
[5]Tyndall Centre for Climate Change Research, School of Environmental Sciences, University
of East Anglia, Norwich, UK.
**1. Abstract**
Atmospheric $CO_2$ levels continue to rise, increasing the risk of severe impacts on the Earth
system, and on the ecosystem services that it provides. Artificial Ocean Alkalization (AOA)
is capable of reducing atmospheric $CO_2$ concentrations, surface warming and addressing
ocean acidification. Here we simulate global and regional responses to alkalinity addition
(0.25 PmolAlk/year) using the CSIRO-Mk3L-COAL Earth System Model in the period
2020-2100, under high (RCP8.5) and low (RCP2.6) emissions. While regionally there are
large changes associated with locations of AOA, globally we see only a very weak
dependence on where and when AOA is applied. We see that under RCP2.6, while the carbon
uptake associated with AOA is only ~60% of the total under RCP8.5, the relative changes in
temperature are larger, as are the changes in pH (1.4x) and aragonite saturation (1.7x). The
results of this modelling study are significant as they demonstrate that AOA is more effective
under lower emissions, and the higher the emissions the more AOA required to achieve the
same reduction in global warming and ocean acidification.  Finally, our simulations show
AOA in the period 2020-2100 is capable of offsetting global warming and ameliorating ocean
acidification increases due to low emissions, but regionally the response is more variable.



### 1. Introduction

Atmospheric carbon dioxide ($CO_2$) levels continue to rise primarily as a result of human
activities. Recent studies have suggested that even deep cuts in emissions may not be
sufficient to avoid severe impacts on the Earth system, and the ecosystem services that it
provides (Gasser et al., 2015). Recent international negotiations (UNFCCC, 2015) agreed to
limit global warming to well below 2°. The application of Carbon Dioxide Removal (CDR),
sometimes referred to as "Negative Emissions", appears to be required to achieve this goal,
as emission reductions alone are likely to be insufficient (Rogelj et al., 2016). In this context,
there is an urgent need to assess how Carbon Dioxide Removal (CDR) could help either
mitigate climate change or even reverse it, and to understand the potential risks and benefits
of different options.

While warming represents a major imminent global threat, including through coral bleaching
which is already significantly impacting the natural environment (Hughes et al., 2017) ocean
acidification poses an additional and equally significant threat to the marine environment.
Ocean acidification occurs as $CO_2$ taken up by the ocean reacts with the seawater to reduce
the carbonate ion concentration and decrease the pH.  Annually the oceans take up about 28%
of anthropogenic $CO_2$ emitted (Le Quéré et al., 2015).

Ocean acidification is the unavoidable consequence of rising atmospheric $CO_2$ levels and will
impact the entire marine ecosystem - from plankton at the base, to fish at the top. Potential
impacts include changes to calcification, fecundity, organism growth and physiology, species
composition and distributions, food web structure and nutrient availability (Doney et al.,
2012;Dore et al., 2009;Fabry et al., 2008;Iglesias-Rodriguez et al., 2008;Munday et al.,
2010;Munday et al., 2009). Within this century, the impacts of ocean acidification will
increase in proportion to emissions (Gattuso et al., 2015). Furthermore, these changes will be
long lasting, persisting for centuries or longer even if emissions were halted (Frolicher and
Joos, 2010).

To date many different CDR techniques have been proposed both on the land and in the
ocean (Royal Society, 2009;National Research Council, 2015). Their primary purpose is to
reduce atmospheric $CO_2$ levels; and most CDR methods will ameliorate the impacts of ocean
acidification,  although some proposed techniques such as ocean pipes (Lovelock and Rapley,

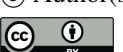



2007) and micro-nutrient addition (Keller et al., 2014) may actually lead to an acceleration of
ocean acidification in surface waters.

Artificial Ocean Alkalization (AOA) through altering the chemistry of seawater both
enhances ocean carbon uptake, thereby reducing atmospheric $CO_2$, while at the same time
directly reversing ocean acidification. AOA can be thought of as a massive acceleration of
the natural processes of chemical weathering of minerals that may have played a role in
modulating the climate on geological timescales (Zeebe, 2012;Colbourn et al., 2015).
Alkalinity changes may also have played an important role in controlling glacial-interglacial
cycles of atmospheric $CO_2$ e.g. Sigman and Boyle (2000).

Specifically, as alkalinity enters the ocean the pH increases leading to elevated carbonate ion
concentration, reduction in hydrogen ion concentration and decrease in the concentration of
aqueous $CO_2$ (or $pCO_2$).  This in turn enhances the disequilibrium of $CO_2$ between the ocean
and atmosphere (or $\Delta pCO_2 = pCO_2^{ocean} - pCO_2^{atmosphere}$) leading to increased ocean carbon
uptake, and reduction in atmospheric $CO_2$ concentration. These increases in pH and
carbonate ion concentration reverse the ocean acidification due to uptake of anthropogenic
$CO_2$.

Kheshgi (1995) first proposed AOA as a method of CDR. Renforth and Henderson (2017)
review the early experimental, engineering and modelling work undertaken to investigate
AOA. From the observational perspective we draw particular attention to the experimental
work of Albright et al. (2016) which provided an in situ demonstration of localised AOA to
offset the observed changes in ocean acidification on the Great Barrier Reef that have
occurred since the pre-industrial period.

Several modelling studies have explored the impacts of AOA both on carbon sequestration
and ocean acidification. Using ocean only biogeochemical models Kohler et al. (2013)
explored AOA via olivine addition. Olivine, in addition to increasing alkalinity also adds iron
and silicic acid, both of which can enhance ocean productivity (Jickells et al.,
2005;Ragueneau et al., 2000). Kohler et al. (2013) estimated the response of atmospheric
$CO_2$ levels and pH to different levels of olivine addition over the period 2000-2010, and later
this was extended to 2100 by Hauck et al. (2016). These studies demonstrate a global impact,
that appeared to scale with the amount of olivine added. Importantly, Kohler et al. (2013)



showed that the global effect of alkalinity added along shipping routes (as an analogue for
practical implementation) was not significantly different from that of alkalinity added in a
highly idealized uniform manner.

Ilyina et al. (2013) explored the potential of AOA to mitigate rising atmospheric $CO_2$ levels
and ocean acidification in ocean-only biogeochemical simulations, and showed that AOA has
the potential to ameliorate future changes due to high emissions. They did not limit the
amount of AOA, as their goal was to offset the projected future changes; and showed that the
amount of AOA required to do this would drive the carbonate system to levels well above
pre-industrial levels. Ilyina et al. (2013) also conclude that local AOA could potentially be
used to offset the impact of ocean acidification, with enhanced $CO_2$ uptake being only a side
benefit. This regional approach was explored further by Feng et al. (2016) who suggested that
local AOA in the tropical ocean, in areas of high coral calcification, has the potential to offset
the impact of future rising atmospheric $CO_2$ levels under a high emissions scenario (RCP8.5).
This study also revealed strong regional sensitivities in the response of ocean acidification
related to the locations in which it was applied.

To date several studies estimate the response of the Earth system to AOA. Gonzalez and
Ilyina (2016) used an Earth System Model (ESM) to estimate the AOA required to reduce
atmospheric concentrations from high emissions scenario (RCP8.5) to the medium emissions
scenario (RCP4.5). They estimated that to mitigate the associated 1.5K warming difference,
via reducing atmospheric $CO_2$ concentrations by ~400 ppm, would require an addition of 114
Pmol of alkalinity (between 2018-2100), and would come at the cost of very large
(unprecedented) changes in ocean chemistry.

Keller et al. (2014) used an Earth System Model of Intermediate Complexity (EMIC) to
explore the impact of AOA over the period 2020-2100, to a globally uniform addition of
alkalinity (0.25 PmolALK/yr), an amount based on the estimated carrying capacity of global
shipping following Kohler et al. (2013). Keller et al. (2014) showed that AOA led to
reduction in atmospheric $CO_2$ of 166 PgC or ~78ppm, a net cooling of 0.26K and a global
increase in ocean pH of 0.06 in the period 2020-2100.

To date not all modelling studies have been emissions driven, and this is important as
potential climate and carbon cycle feedbacks may not have been accounted for. Capturing



these feedbacks is critical as they have the potential to be very large (Jones et al., 2016).
Further, no studies have explored the impact of AOA under low emissions scenarios such as
RCP2.6. This is important because scenarios that limit warming to 2° or less, currently utilize
considerable land based CDR via afforestation and/or Bio-Energy with Carbon Capture and
Storage (BECCS) the feasibility of which are increasingly questioned due in part to limited
land (Smith et al., 2016), whereas the potential CDR capacity of the oceans is orders of
magnitude greater (Scott et al., 2015).

In this work, using a fully coupled Earth System Model (CSIRO-Mk3L-COAL), which
includes climate and carbon feedbacks, we investigate the impact of AOA on ocean
acidification, land and ocean carbon uptake and warming. Specifically, the question this
study tackles is: What is the impact of global and regional AOA on the Earth System, and
how efficient it is at mitigating global warming and ocean acidification under high and low
emissions trajectories?

**2. Methods**
*2.1 Model Description*
The model simulations were performed using the CSIRO-Mk3L-COAL (Carbon Ocean,
Atmosphere, Land) Earth System Model which includes climate-carbon interactions and
feedbacks. (Matear and Lenton, 2014;Zhang et al., 2014a). The ocean component of the
Earth System Model has a resolution of 2.8° by 1.6° with 21 vertical levels. The ocean
biogeochemistry is based on (Lenton and Matear, 2007;Matear and Hirst, 2003) simulating
the distributions of phosphate, oxygen, dissolved inorganic carbon and alkalinity in the
ocean. The  model simulates particulate inorganic carbon (PIC) production as function of
particulate organic carbon (POC) production  via the rain ratio (9%) following (Yamanaka
and Tajika, 1996).  This ocean biogeochemical model was shown to simulate the observed
distributions of total carbon and alkalinity in the ocean (Matear and Lenton, 2014) and
phosphate (Duteil et al., 2012).

The atmosphere resolution is 5.6° x 3.2° with 18 vertical layers. The land surface scheme
uses CABLE (Best et al., 2015) coupled to CASA-CNP (Wang et al., 2010;Mao et al., 2011)
which simulates biogeochemical cycles of carbon, nitrogen and phosphorus in plants and
soils. The response of the land carbon cycle was shown to realistically simulate the observed
biogeochemical fluxes and pools on the land surface (Wang et al., 2010).




To quantify the changes in ocean acidification we calculate pH changes on the total scale
following the recommendation of Riebesell et al. (2010). To calculate the changes of
carbonate saturation state we use the equation of Mucci (1983).

*2.2 Model Experimental Design*
Our ESM was spun-up under a preindustrial atmospheric $CO_2$ concentration of 284.7 ppm,
until the simulated climate was stable (> 2000 years) (Phipps et al., 2012). From the spun-up
initial climate state, the historical simulation (1850 - 2005) was performed using the
historical atmospheric $CO_2$ concentrations as prescribed by the CMIP5 simulation protocol
(Taylor et al., 2012).

Following the historical concentration pathway from 2006, two different future projections to
2100 were made using the atmospheric $CO_2$ emissions corresponding to Representative
Concentration Pathways of low emissions (RCP2.6) and high emissions (RCP8.5 or 'business
as usual') (Taylor et al., 2012). All simulations include the forcing due to non-$CO_2$
greenhouse gas concentrations (Taylor et al., 2012). We define RCP8.5 and RCP2.6 as our
control cases for the corresponding experiments below.

In the period 2020-2100 we undertook a number of AOA experiments using a fixed quantity
of 0.25 Pmol/yr of alkalinity, the same amount used by Keller et al. (2014). Consistent with
this study we applied AOA in the surface ocean all year-round in ice-free regions, set to be
between 60°S and 70°N. For each of the two emissions scenarios, we considered four
different regional applications of AOA, shown in Figure 1. These are: (i) AOA globally
(AOA_G) between 60S and 70N; (ii) the higher latitudes comprising the subpolar northern
hemisphere oceans (40N -70N) and the (ice-free) Southern Ocean (40S-60S) (AOA_SP); (iii)
the subtropical oceans (15-40N and 15S and 40S; AOA_ST); and (iv) in the equatorial
regions (15N-15S; AOA_T). In this study, we only look at the response of the Earth system
to alkalinity injection. We do not consider the biogeochemical response to other minerals and
elements associated with the proposed sourcing of alkalinity from the application of finely
ground ultra-mafic rocks such as olivine and fosterite, nor dissolution processes required to
increase alkalinity e.g. Montserrat et al. (2017).

**3. Results and Discussion**



To aid in presenting our results and to compare these with previous studies we first discuss
the carbon cycle, global surface warming (2m surface air temperature), and response and
ocean acidification response to the 4 different AOA experiments under the high (RCP8.5)
and low (RCP2.6) emissions scenarios. We then look at the regional behaviour of the
simulations in the different AOA experiments.

*3.1 Global Response*
For each emission scenario, we simulated 4 different AOA experiments, which all had the
same 0.25 Pmol/yr of alkalinity added. As anticipated by 2100, AOA increased the global
mean surface ocean alkalinity relative to the corresponding scenario control case, with the
magnitude of the increase in alkalinity being dependent on where it was added (Table 1).
Polar addition (AOA_SP) led to the smallest net increase in surface alkalinity, while tropical
addition (AOA_T) produced the greatest increase. As expected, the global mean changes in
surface alkalinity between emissions scenarios are very small (less than 3 µmol/kg
difference). The slightly greater increase in surface values in alkalinity under RCP8.5 likely
reflects enhanced ocean stratification under higher emissions (Yool et al., 2015).

|              | RCP8.5 | RCP2.6 |
|--------------|--------|--------|
| AOA_G -RCP   | 108.3  | 105.1  |
| AOA_SP-RCP   | 79.7   | 74.4   |
| AOA_ST-RCP   | 115.1  | 112.9  |
| AOA_T-RCP    | 129.8  | 127.1  |


*Table 1 For the two RCP scenarios the relative increase in ocean surface alkalinity*
*(µmol/kg) between each AOA experiment and control experiment in 2100.*

*3.1.1 Carbon Cycle*

The large atmospheric $CO_2$ concentration at the end of the century under RCP8.5 reflects the
large increase in emissions projected (under RCP8.5), while under RCP2.6 a similar
atmospheric concentration of $CO_2$ is seen 2100 as at the beginning of the simulation (2020)
(Figure2a).  We note that atmospheric $CO_2$ levels in our CSIRO-MK3L-COAL for the
control cases are greater than for their respective concentration driven RCPs due to nutrient
limitation in the land, leading to reduced carbon uptake (Zhang et al., 2014a) .




Under all emissions scenarios and experiments AOA leads to reduced atmospheric $CO_2$
levels (Figure 2a). Under RCP8.5, AOA reduces atmospheric concentration by 82-86 ppm,
this represents ~16% decrease in atmospheric concentration (525 ppm increase over the
period 2020-2100). In contrast to RCP8.5, AOA under RCP2.6 leads to a smaller reduction in
atmospheric concentration (53-58 ppm).  Figure 2a shows that by the of the century AOA
more than compensates for the projected increase in atmospheric $CO_2$ due to RCP2.6.

Over the 2020-2100 period, the reduction in atmospheric $CO_2$ levels associated with AOA is
primarily due to increased ocean carbon uptake, offset by small decreases in the land surface
carbon uptake (Table 2). In the ocean, RCP8.5 has much greater net uptake than RCP2.6,
about 1.5 times more, due to the larger (and growing) disequilibrium between the atmosphere
and ocean.

In the ocean, the relative increase in carbon uptake in response to AOA is primarily abiotic in
nature. Consistent with Keller et al. (2014) and Hauck et al. (2016) the simulated changes in
ocean export production were very small (~0.2 PgC) under RCP8.5. While under RCP2.6 it
was slightly larger at 1.2 PgC, but still less than 1% percent of the total ocean increase
simulated under AOA. In contrast, the relative decreases in land carbon uptake were biotic in
nature. The simulated cooling over land drove a reduction in net primary production that
more than offset the decrease in carbon flux due in heterotrophic respiration. On the land, the
RCP8.5 simulated a smaller reduction in carbon uptake than RCP2.6, potentially due to a
smaller relative cooling (Table 3).

For both emissions scenarios, the 4 AOA experiments all produced similar reductions in
atmospheric $CO_2$ concentrations (Figure 2) with less than a 5% difference in the total land
and ocean carbon uptake. The global changes in land and ocean carbon uptake do not appear
to be very sensitive to where we add the alkalinity to the surface ocean. This is consistent
with Kohler et al. (2013) who saw little difference in adding olivine along existing shipping
tracks, versus uniformly adding it to the surface ocean. It is also consistent with regional
addition studies Ilyina et al. (2013) and Feng et al. (2016) who demonstrated a global impact.

Our simulated total increased carbon uptake under AOA_G with RCP8.5 (179 PgC) is
comparable to the 166 PgC reported by (Keller et al., 2014). Their cumulative increase in





ocean carbon uptake by 2100 of 181 PgC was in very good agreement with our value of 184
PgC. However, they simulated a reduction in land uptake nearly twice the -5.8 PgC reduction
in our AOA_G simulation. These differences may reflect both the sensitivity of the simulated
climate feedbacks, and differences in land surface models.

| | Total RCP8.5 | Ocean RCP8.5 | Land RCP8.5 | Total RCP 2.6 | Ocean RCP 2.6 | Land RCP 2.6 |
|---|---|---|---|---|---|---|
| AOA_G-RCP | 178.6 | 184.4 | -5.8 | 121.1 | 143.1 | -22.1 |
| AOA_SP-RCP | 183.3 | 188.1 | -4.8 | 122.1 | 145.2 | -24.1 |
| AOA_ST-RCP | 180.7 | 185.1 | -4.4 | 122.0 | 143.1 | -21.2 |
| AOA_T-RCP | 174.5 | 177.2 | -2.7 | 116.0 | 139.2 | -23.1 |


*Table 2 The total integrated additional carbon uptake (in PgC) in the period 2020-2100 in*
*different experiment and emissions scenarios, negative denotes enhanced uptake.*

*3.1.2 Surface Air Temperature*
In the control simulations, the global mean surface air temperature (SAT; 2m) increased in
the period 2020-2100 with RCP2.6 simulating a net warming of 0.4±0.1K while RCP8.5
warmed by 2.7±0.1K (2081-2100). All AOA experiments simulated a reduction in global
mean SAT relative to their corresponding control simulation (Figure 2b). Within each
emissions scenario the global mean SAT decline associated with AOA is always greater and
more variable over the land than ocean (Table 3). While the mean cooling, in the period
2081-2100, is also greater over the land under RCP8.5 than RCP2.6, potentially reflecting
feedbacks such as  soil-moisture (Seneviratne et al., 2010) snow and ice cover changes.
However, these changes are associated with large interannual large variability, and therefore
not significantly different.

Under RCP2.6, all the AOA experiments keep warming levels much close to values in 2020
than RCP2.6 by the end of this century (2100; Figure 2b). In contrast, under the RCP8.5
scenario, none of the AOA experiments have a significant impact on the projected warming



by the end of this century (less than 10%) reflecting the large warming anticipated under high
emissions (Rogelj et al., 2012).

|  | Total RCP8.5 | Ocean RCP8.5 | Land RCP8.5 | Total RCP 2.6 | Ocean RCP 2.6 | Land RCP 2.6 |
|---|---|---|---|---|---|---|
| AOA_G-RCP | -0.16±0.08 | -0.14±0.07 | -0.22±0.15 | -0.25±0.08 | -0.19±0.05 | -0.39±0.22 |
| AOA_SP-RCP | -0.13±0.10 | -0.11±0.07 | -0.18±0.20 | -0.23±0.08 | -0.18±0.05 | -0.35±0.22 |
| AOA_ST-RCP | -0.08±0.05 | -0.06+0.03 | -0.13±0.14 | -0.20±0.09 | -0.15±0.06 | -0.30±0.20 |
| AOA_T-RCP | -0.14±0.06 | -0.12±0.05 | -0.19±0.11 | -0.16±0.06 | -0.13±0.05 | -0.24±0.16 |


*Table 3 The differences in global mean surface air temperature and their standard deviations*
*(1-$\sigma$) (K; SAT; 2m) for 2090 (in the period 2081-2100) for different AOA experiments and*
*emissions scenarios relative to the emissions scenarios with no AOA.*

Within each of the scenarios, there is some variability in the magnitude of the cooling within
the 4 different AOA experiments, however, these differences are smaller than the interannual
variability over the last 2 decades of the simulations. Therefore, it appears that the global
mean SAT decline with AOA is not very sensitive to where the alkalinity is added under
either emission scenario.

The global mean cooling associated with AOA_G under RCP8.5 (-0.16±0.08K; 2081:2100)
is close to the mean surface air temperature decrease of -0.26K reported by (Keller et al.,
2014) for the same levels of AOA. These differences may reflect the simplified atmospheric
representation of the UVIC Intermediate Complexity Model and different climate
sensitivities.

*3.1.3 Ocean Acidification*

Here we quantify changes in ocean acidification in terms of pH and aragonite saturation state
changes.   We consider these two diagnostics because they are associated with different





biological impacts and not necessarily well correlated (Lenton et al., 2016). In the future, the
global mean changes in pH and aragonite saturation state will be proportional to the
emissions trajectories following Gattuso et al. (2015) with the largest changes associated with
the higher emissions (RCP8.5) (Figure 2c-d). By 2100 despite a return to 2020 values of
atmospheric $CO_2$ concentration under RCP2.6 (Figure 2), neither pH or aragonite saturation
state return to 2020 values, consistent with Mathesius et al. (2015).

In the 2020-2100 period, AOA under RCP2.6 led to much larger increases in surface pH and
aragonite saturation state, more than 1.3 times, and more than 1.7 times that of RCP8.5
respectively (Table 4). These changes reflect the differences in the mean state associated with
high and low emissions, specifically the difference between Alkalinity and DIC. The values
of DIC in the upper ocean are larger under RCP8.5 than RCP2.6, and therefore the ALK-DIC
is higher.  For a given addition of alkalinity, the increase in the upper ocean DIC will be
greater in the high emission case than the low emission case due to the Revelle Factor
(Revelle and Suess, 1957).  Consequently, the difference between Alkalinity and DIC with
AOA increases less in the high emission scenario than the low scenario, which translates into
smaller increases in pH and aragonite saturation state in the high scenario.

While there was a significant difference in pH and aragonite saturation state changes with
AOA between high and low emissions cases, the global mean changes for different AOA
experiments within each scenario is quite similar (Table 4).  The exception being the
AOA_SP experiment, where its pH and aragonite saturation state changes are only ~75% of
the change in the other AOA experiments. This reduced change in the polar region is
consistent with the smaller changes in the surface ocean alkalinity values associated with
AOA_SP (Table 1). These differences at higher latitudes reflect the enhanced subduction of
alkalinity away from the surface ocean into the ocean interior that occurs in the high latitude
oceans (Groeskamp et al., 2016).

AOA_G under RCP8.5 leads to a relative increase in pH of 0.06 which is consistent with
(Keller et al., 2014), while our relative increase in aragonite saturation state (0.28) is also
very close to their simulated value (0.31). To put these changes into context, the estimated
decrease in pH since the preindustrial is 0.1 units (Raven et al., 2005).





| | Aragonite RCP8.5 | pH RCP8.5 | Aragonite RCP2.6 | pH RCP 2.6 |
|---|---|---|---|---|
| AOA_G-RCP | 0.28 | 0.06 | 0.50 | 0.07 |
| AOA_SP-RCP | 0.20 | 0.05 | 0.39 | 0.07 |
| AOA_ST-RCP | 0.30 | 0.06 | 0.54 | 0.08 |
| AOA_T-RCP | 0.28 | 0.06 | 0.5 | 0.07 |


*Table 4 The relative differences in surface value of aragonite saturation state and pH*
*between the AOA experiments and the high and low emissions scenarios in 2100*

*3.2 Regional Responses*
For both RCP scenarios, there are large regional differences in the relative surface changes in
alkalinity, temperature, and ocean acidification associated with the different AOA
experiments.  The regional nature of these changes is closely associated with where alkalinity
addition is applied, and the two different emissions scenarios considered here do not differ
significantly in their behaviour. This implies that any differences in stratification and
overturning circulation between the two scenarios are insufficient to significantly modulate
the response to AOA.

*3.2.1. Surface Alkalinity*

For both scenarios, the greatest surface alkalinity changes occur where the alkalinity is added
(Figure 3). Spatially under either emission scenario, the relative differences in 2090 are very
similar, consequently we only show the changes under RCP2.6 (Figure 3). The only
significant differences occur in the Arctic, reflecting larger longer-term changes in alkalinity
projected under higher emissions (Yamamoto et al., 2012).

Overall the greatest increases are seen in the tropical ocean (AOA_T) suggesting this is the
most efficient region in retaining the added alkalinity in the upper ocean.  This reflects the
fact that subduction processes in the tropical ocean are less efficient than other regions such
as the higher latitudes. In the (ice-free) subpolar oceans (AOA_SP) produced the smallest
relative increase in alkalinity, and this reflects the strong and efficient surface to interior





connections occurring at higher latitudes (Groeskamp et al., 2016). The global mean relative
increase associated with AOA in the subtropical gyres (AOA_ST) and globally (AOA_G) fall
between the tropical and higher latitude values. In the case of AOA_ST this reflects the time-
scales associated with circulation of the subtropical gyres.

The most modest relative increase in alkalinity occurs in the non-ice-free regions where
alkalinity is not explicitly added. Interestingly even when alkalinity is added in the very high
latitude Southern Ocean it is carried northward by the Ekman current explaining the very
modest increase in the region where AOA occurs between 50S to 60S. In terms of the total
alkalinity added to the surface ocean, about one-third remains in the upper 200m in 2100.
Specifically, for AOA_G we see 31% remains, AOA_T and AOA_ST: 34%, while for
AOA_SP: 22-24% remains, which (as anticipated) is lower than in other regions.

Spatially AOA in the higher latitude regions (AOA_SP) leads to very large relative increases
in alkalinity (> 1000 μmol/kg; 2090) occurring along the northern most boundary of the
Northern Subpolar Gyres, particularly the North Pacific. Clearly in this region the rate of
AOA exceeds the rate of subduction allowing alkalinity to build up. Large relative increases
in alkalinity also occur in the Southern Ocean under AOA_SP, particularly along Western
Boundary Currents. However, in contrast to northern high latitudes the values still remain
quite low suggesting that the rate of addition does not exceed the rate of subduction even
under the highest emission scenario.

AOA_ST shows a large relative increase of ~300 μmol/kg (2081-2100) in the subpolar gyre
regions. Overall, we find that these relative increases are quite homogenous across the entire
subtropical gyres, with strong mixing with tropical waters leading to significant relative
increases in tropical Atlantic, Western Pacific and Indian Oceans.  Within the tropical ocean,
under AOA_T the largest relative changes are found across the entire tropical Indian Ocean
(~ 400 μmol/kg) with large relative increases also seen the Indonesian seas (~280 μmol/kg;
2081-2100). Away from the tropical Indian Ocean we find that relatively homogenous
increases occur in the Western Pacific and the Atlantic, with much more modest relative
increases in the Eastern Pacific reflecting the dominant East to West upper ocean circulation.
Consistent with the response of AOA_ST, AOA_T leads to relative increases in surface
alkalinity in the AOA_ST region of ~130 μmol/kg (2081-2100).




In the case of AOA_G a relatively uniform net increase in alkalinity occurs in all regions
with the exception of the upwelling regions such as the tropical Pacific, which showed a
more modest relative increase. In AOA_G there is little evidence of any of the very large
increases in alkalinity seen in the more regional AOA experiments. This spatial pattern of
relative increase is broadly consistent with the pattern of global alkalinity increase simulated
by Ilyina et al. (2013) and Keller et al (2014) for AOA in the (ice-free) global ocean.
*3.2.2 Changes in the interior distribution of alkalinity in the global ocean*
As only about 30% of the total AOA remains in the upper 200m, we explore the fate of this
alkalinity in the interior ocean in the zonal sections of alkalinity (Figure 4). As the pattern is
very similar, between RCP2.6 and RCP8.5 we only show RCP2.6, noting that in the North
Atlantic where the projected ocean stratification is stronger under higher emissions (not
shown) leading to slightly decreased subsurface values. This increased stratification is
consistent with other studies e.g. (Yool et al., 2015).

Unlike the surface plots of AOA, the relative increases in alkalinity due to AOA are very
similar across all experiments. This heterogeneous pattern of alkalinity increase is associated
with water entering the interior ocean along specific surface to interior pathways (Groeskamp
et al., 2016). Specifically, we see alkalinity moving into the interior ocean along the poleward
boundaries of the subtropical gyres, associated with the formation and subduction of mode
waters, and an increase in the subtropical gyres associated with large-scale downwelling, and
deep mixing in the North Atlantic. The changes in alkalinity are mainly found in the upper
ocean (<1000 m) which reflects the relatively short period of alkalinity addition. Given the
short period this is analogous to present-day observed distributions of anthropogenic carbon
(Sabine et al., 2004).

As the changes in export production are very small, the large changes in the interior alkalinity
concentrations primarily reflect the physical transport, rather than the sinking and
remineralization of calcium carbonate.  Clearly other biological processes, not represented in
our model, have the potential to impact the surface and interior values of alkalinity (Matear
and Lenton, 2014). One such process is the reduction in ratio of PIC:POC under higher
emissions (Riebesell et al., 2000) however it has been shown that even a very large reduction



in PIC production (50%) would not significantly impact our results (Heinze, 2004)
Unfortunately, at present, the magnitude and sign of many of these other feedbacks remains
poorly known (Matear and Lenton, 2014) consequently quantifying their impact on our
results is very difficult, and beyond the scope of this study.

*3.2.3 Ocean Carbon Cycle Response*

The similarity in global ocean carbon uptake associated with all AOA experiments for a
given emission scenario hides the large spatial differences between simulations. Given that
the largest carbon cycle response occurs in the ocean (Table 2), we focus on this response for
RCP8.5 and RCP2.6 (Figures 5 and 6).  As expected ocean carbon uptake is strongly
enhanced in the regions of AOA. Away from regions of AOA there is a reduction in carbon
uptake, associated with the weakening of the gradient in $CO_2$ between the atmosphere and
ocean due to AOA. Interestingly the largest increase spatially occurs in the Southern Ocean
under AOA_SP for RCP2.6, while in contrast the largest changes under RCP8.5 occurs in the
tropical ocean under AOA_SP. The very small changes in export production in RCP2.6 were
located in the Arabian Sea (not shown), and while these changes are < 1% of the total change
in carbon uptake, nevertheless they maybe important regionally.

*3.2.4 Temperature (SAT)*

The decrease in global mean SAT associated with all AOA experiments for a given emission
scenario again hides the large spatial differences between the simulations.  The response of
surface temperature is spatially very heterogeneous (Figures 7 and 8), and while the regional
surface temperature changes are very similar between the two emissions scenarios. The
exception to this is the Arctic which did not show a consistent response across the different
AOA experiments. Under both emission scenarios, the largest cooling associated with AOA
occurs over Northern Russia and Canada, and Antarctica (greater than a -1.5K cooling) with
a larger cooling in these regions under RCP2.6.

In the surface ocean, AOA in the RCP2.6 scenario shows a net cooling over the ocean, with
the exception of the North Atlantic, east of New Zealand, and off the southern coast of
Alaska which show very a modest warming. A similar pattern is evident in RCP8.5 however





there is a greater cooling in the high latitudes, and less cooling in the lower latitudes than
under RCP2.6.

*3.2.5 Ocean Acidification Response*

Globally the response of pH and aragonite saturation state associated with AOA are similar,
however large spatial and regional differences are present.  To aid in the interpretation of
changes in aragonite saturation state, overlain on the aragonite saturation state maps (Figures
9 and 10) are the contours corresponding to the value of 3, the approximate threshold for
suitable coral habitat (Hoegh-Guldberg et al., 2007). On these surface maps and subsequent
section plots (Figures 13  and 14) we plot the saturation horizon i.e. the contour
corresponding to the transition from chemically stable to unstable (or corrosive), i.e.
aragonite saturation state is equal to 1 (Orr et al., 2005).

The largest relative changes in pH and aragonite saturation state were associated with regions
of AOA, reflecting increases in the surface values of alkalinity. All simulations increase pH
and aragonite saturation state in the Arctic despite no direct addition in this region, with the
largest changes here associated with AOA_G and AOA_SP. Interestingly all simulations
show little to no increase in the high latitude Southern Ocean, consistent with more efficient
transport of the added alkalinity into the ocean interior.

The changes in pH associated with AOA experiments under RCP8.5 while spatially very
different, particularly when added in the subpolar ocean, are still much less than the
decreases associated with RCP8.5 with no AOA. In terms of aragonite saturation state the
conditions for coral growth in the tropical ocean remain very unfavorable by the end of
century (i.e. aragonite saturation state < 3) under all regional and global experiments, with
the exception of AOA_T, where only a very small region in the Central Pacific Ocean
exhibits suitable conditions.

Consistent with Feng et al. (2016) we find that this level of AOA under RCP8.5 is
insufficient to ameliorate or significantly alter the large-scale changes in ocean acidification.
More positively, at the higher latitudes the saturation horizon is moved poleward with the
largest shift associated with AOA_SP, and the smallest shift at the high latitudes occurring
under AOA_T. Consistent with these changes we see a deepening of the saturation horizon



everywhere (Figure 13), and little difference spatially between AOA experiments, consistent
with zonal mean changes in alkalinity for the 4 AOA experiments.

The spatial pattern of changes associated with AOA under RCP2.6 are broadly consistent
with those seen under higher emissions, however the magnitude of the response is much
larger again due to the larger differences between Alkalinity and DIC with AOA under
RCP2.6. In terms of aragonite saturation state, the area of tropical ocean favourable for corals
is considerably expanded suggesting that conditions for tropical coral growth are improved
under AOA. As anticipated the largest changes in the area favourable for tropical corals is
associated with AOA_T, closely followed by AOA_ST.  As the saturation horizon does not
reach the surface under RCP2.6 we can only look at the changes in the interior ocean. Here
there is a deepening in the saturation horizon in all experiments of a very similar magnitude
(Figure 14), with the exception of the Arctic. Here the response of the saturation horizon is
more sensitive to the location of the AOA varying between ~100m under AOA_T and ~280
m under AOA_SP.

Spatially the large changes in ocean acidification in response to AOA under RCP2.6 more
than compensate for the changes in ocean chemistry due to low emissions in the period 2020-
2100. Globally, in the changes in the period 2020-2100 are sufficient to reversed or
compensate the changes since the preindustrial (1850).  However spatially in some regions
such as equatorial upwelling, an important area of global fisheries (Chavez et al., 2003),
AOA in fact leads to higher values of aragonite saturation state and pH than the ocean
experienced in preindustrial period (Feely et al., 2009).  We can only speculate on the
potential impact on marine biota through a reduction in aqueous $CO_2$ and elevated pH levels
in these regions. For  a recent review of the potential impact of rising pH and Aragonite
saturation state on marine organisms we direct the reader to Renforth and Henderson (2017).

*3.2.6 Importance of Seasonality*

In this paper, while we have focused on year-round AOA, as a sensitivity experiment we also
explored whether AOA added in summer or winter was more efficient. To do this we focused
on the higher latitudes regions where the largest seasonal changes in mixing are found (de
Boyer Montegut et al., 2004;Trull et al., 2001).  Here we tested whether AOA in either
summer or winter was more effective than year-round addition. To test this for RCP8.5 we



add alkalinity only during the summer at half of the annual rate (or 0.125PmolALK/year) in
the AOA_SP region.

Our results showed that the response to AOA in summer was very close to 50% of the
response of the year-round addition associated with AOA_SP (or 0.25PmolALK/year). This
suggests that the response of AOA appears invariant to when the alkalinity is added. This
also suggests, consistent with published studies e.g. (Keller et al., 2014;Feng et al.,
2016;Kohler et al., 2013) that the response of the ocean to different quantities of AOA is
scalable under the same emissions scenario. Whether this is true under very much larger
additions of alkalinity as simulated by (Gonzalez and Ilyina, 2016) is less clear.

**4.  Summary and Concluding Remarks**

Integrated Assessment Modelling for the Intergovernmental Panel on Climate Change shows
that $CO_2$ removal (CDR) may be required to achieve the goal of limiting warming to well
below 2° (COP21) (Fuss et al., 2014). Of the many schemes that have been proposed to limit
warming, only Artificial Ocean Alkalization (AOA) is capable of both reducing the rate and
magnitude of global warming through reducing atmospheric $CO_2$ concentrations, while
simultaneously directly addressing ocean acidification. Ocean acidification, while receiving
often less attention, is likely to have very long lasting and damaging impacts on the entire
marine ecosystem, and the ecosystem services it provides.

Here, for the first time we investigate the response of a fully coupled climate ESM, i.e. that
accounts for climate-carbon feedbacks, under high (RCP8.5) and low (RCP2.6) emissions
scenarios to a fixed addition of alkalinity (0.25PmolALK/year). We explore the effect of
global and regional application of AOA focusing on the subpolar gyres, the subtropical gyres
and the tropical ocean. To assess AOA, we look at changes in surface air temperature, carbon
cycling and ocean acidification (aragonite saturation state and pH) in the period 2020-2100.

Consistent with other published studies we see that AOA leads to reduced atmospheric $CO_2$
concentrations, cooler global mean surface temperatures, and reduced levels of ocean
acidification.  Globally for these metrics we observed that they do not vary significantly
between the various AOA experiments under each emissions scenario. This implies that at
this scale there is little sensitivity of the global responses to the region where AOA is applied.



We also investigate as a sensitivity experiment adding alkalinity in different seasons and see
little difference in response to when AOA was undertaken.

We see under AOA that the increased carbon uptake is dominated by the ocean.  Under
RCP8.5 the changes due to AOA are only capable of reducing atmospheric concentrations by
a maximum of 86 ppm versus the projected change of 560ppm, and as such the response of
the climate system remains strongly dominated by warming. This is consistent with published
studies of the response of the climate system under RCP8.5, and studies that have estimated
the amount of AOA required to counteract a high emissions trajectory.

In contrast AOA under RCP2.6 while only capable of reducing atmospheric $CO_2$ levels by 58
ppm, is sufficient to reduce atmospheric $CO_2$ concentrations and warming to close to 2020
levels at the end of the century. This is significant as it suggests that in combination with
rapid reduction in emissions, AOA could make an important contribution to the goal to keep
global mean temperatures below 2°. However, AOA under RCP2.6 does not ameliorate
spatial changes of carbon uptake associated with RCP2.6, resulting in a reduced uptake in the
terrestrial biosphere and increased uptake in the ocean. This highlights that while the
atmospheric $CO_2$ and warming may be reversible, the response of individual components of
the Earth System to different CDR may not be (Lenton et al., 2017).

Interestingly despite the impact of AOA on the atmospheric $CO_2$ concentration under RCP2.6
being only ~60% of the value the under RCP8.5, we see much larger changes in ocean
acidification associated with RCP2.6 than RCP8.5, more than 1.3 times in pH and more than
1.7 times in aragonite saturation state. This reflect the larger reductions of the difference
between ALK and DIC that occurs under RCP2.6.  We also see larger relative decreases in
global temperature associated with RCP2.6. These results are very important as they
demonstrate that that AOA is more effective under lower emissions in reducing ocean
acidification and global warming.

While there is little sensitivity in the global responses to the region in which AOA is applied,
spatially the largest changes in ocean acidification (and ocean carbon uptake) were seen in
the regions where AOA was applied.  Despite large changes regionally these cannot
compensate for the large changes associated with RCP8.5. Even targeted AOA in the tropical
ocean can preserve only a tiny area of the ocean conducive to healthy coral growth; and even



then the concomitant large warming is likely to be a stronger influence on coral growth than
ocean chemistry (D'Olivo and McCulloch, 2017).

In contrast AOA under RCP2.6 is more than capable of ameliorating the projected ocean
acidification changes in the period 2020-2100. We see that in all cases the area of the tropical
ocean suitable for healthy coral growth expands, with the largest changes are associated with
tropical addition (AOA_T). In some areas, such as the equatorial Pacific, the changes that
have occurred since the preindustrial (1850) are also completely compensated, and in some
cases leads to values that are higher than were experienced in the preindustrial period.

While the amount of alkalinity added in this study is small in comparison to other published
studies, the challenge of achieving even this level of AOA should not be underestimated.
Indeed, it is not clear whether such an effort is even feasible given the cost, logistical,
political and engineering challenges of producing and distributing such large quantities of
alkaline material (Renforth and Henderson, 2017). In the case of RCP8.5 it is unlikely that
this level of AOA could be justified given our results. If emissions can be reduced along an
RCP2.6 type trajectory this study suggests that AOA is much more effective and may provide
a method to remove atmospheric $CO_2$ to complement mitigation, albeit with some side-
effects, and an alternative to reliance on land based CDR.

In this work, and other published studies to date, we have not accounted for role of the
mesoscale in AOA. In the real ocean (mesoscale) eddies are ubiquitous, and associated with
strong convergent and divergent flows, and mixing that plays an important role in ocean
transport (Zhang et al., 2014b). It is plausible that the mesoscale, and indeed fine scale
circulation in the coastal environment e.g. (Mongin et al., 2016a;Mongin et al., 2016b) would
modulate the response to AOA and therefore needs to be considered in future studies.

Furthermore, this is a single model study, and the results of this work need to be tested and
compared in other models. The Carbon Dioxide Removal Model Intercomparison Project
(CDR-MIP) was created to coordinate and advance the understanding of CDR in the earth
system (Lenton et al., 2017). CDR-MIP brings together Earth System models of varying
complexity in a series of coordinated multi-model experiments, one of which is a global
AOA experiment (C4) (Keller et al., 2017). This will allow the response of the earth system
to AOA to be further explored and quantified in a robust multi-model framework, and will

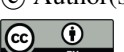



examine important further questions such as including cessation effects of alkalinity addition,
and the long-term fate of additional alkalinity in the ocean.  In parallel, more process and
observational studies e.g. mesocosm experiments, are needed to better understand the
implications of AOA.

**5.  Acknowledgments**
D. P. Keller acknowledges funding received from the German Research Foundation's Priority
Program 1689 "Climate Engineering" (project CDR-MIA; KE 2149/2-1). The authors declare
that there are no conflicts of interest. The model code, simulations and scripts used in this
study are available by contacting Andrew Lenton (andrew.lenton@csiro.au).   We wish to
also to thank Tom W. Trull for his helpful comments on this manuscript.





**6. Biblography**
Albright, R., Caldeira, L., Hosfelt, J., Kwiatkowski, L., Maclaren, J. K., Mason, B. M.,
Nebuchina, Y., Ninokawa, A., Pongratz, J., Ricke, K. L., Rivlin, T., Schneider, K., Sesboue,
M., Shamberger, K., Silverman, J., Wolfe, K., Zhu, K., and Caldeira, K.: Reversal of ocean
acidification enhances net coral reef calcification, Nature, 531, 362-+, 10.1038/nature17155,
664    2016.
Best, M. J., Abramowitz, G., Johnson, H. R., Pitman, A. J., Balsamo, G., Boone, A., Cuntz,
M., Decharme, B., Dirmeyer, P. A., Dong, J., Ek, M., Guo, Z., Haverd, V., Van den Hurk, B.
J. J., Nearing, G. S., Pak, B., Peters-Lidard, C., Santanello, J. A., Stevens, L., and Vuichard,
N.: The Plumbing of Land Surface Models: Benchmarking Model Performance, J
Hydrometeorol, 16, 1425-1442, 10.1175/JHM-D-14-0158.1, 2015.
Chavez, F. P., Ryan, J., Lluch-Cota, S. E., and Niquen, M.: From anchovies to sardines and
back: Multidecadal change in the Pacific Ocean, Science, 299, 217-221, Doi
10.1126/Science.1075880, 2003.
Colbourn, G., Ridgwell, A., and Lenton, T. M.: The time scale of the silicate weathering
negative feedback on atmospheric CO2, Global Biogeochemical Cycles, 29, 583-596,
10.1002/2014GB005054, 2015.
D'Olivo, J. P., and McCulloch, M. T.: Response of coral calcification and calcifying fluid
composition to thermally induced bleaching stress, Scientific reports, 7, Artn 2207
10.1038/S41598-017-02306-X, 2017.
de Boyer Montegut, C., Madec, G., Fischer, A. S., Lazar, A., and Iudicone, D.: Mixed layer
depth over the global ocean: An examination of profile data and a profile-based climatology,
J Geophys Res-Oceans, 109, Artn C12003
10.1029/2004jc002378, 2004.
Doney, S. C., Ruckelshaus, M., Duffy, J. E., Barry, J. P., Chan, F., English, C. A., Galindo,
H. M., Grebmeier, J. M., Hollowed, A. B., Knowlton, N., Polovina, J., Rabalais, N. N.,
Sydeman, W. J., and Talley, L. D.: Climate Change Impacts on Marine Ecosystems, Annu
Rev Mar Sci, 4, 11-37, Doi 10.1146/Annurev-Marine-041911-111611, 2012.
Dore, J. E., Lukas, R., Sadler, D. W., Church, M. J., and Karl, D. M.: Physical and
biogeochemical modulation of ocean acidification in the central North Pacific, P Natl Acad
Sci USA, 106, 12235-12240, Doi 10.1073/Pnas.0906044106, 2009.
Duteil, O., Koeve, W., Oschlies, A., Aumont, O., Bianchi, D., Bopp, L., Galbraith, E.,
Matear, R., Moore, J. K., Sarmiento, J. L., and Segschneider, J.: Preformed and regenerated
phosphate in ocean general circulation models: can right total concentrations be wrong?,
Biogeosciences, 9, 1797-1807, 10.5194/bg-9-1797-2012, 2012.
Fabry, V. J., Seibel, B. A., Feely, R. A., and Orr, J. C.: Impacts of ocean acidification on
marine fauna and ecosystem processes, Ices J Mar Sci, 65, 414-432, Doi
10.1093/Icesjms/Fsn048, 2008.
Feely, R. A., Doney, S. C., and Cooley, S. R.: Ocean Acidification: Present Conditions and
Future Changes in a High-CO2 World, Oceanography, 22, 36-47, Doi
10.5670/Oceanog.2009.95, 2009.
Feng, E. Y., Keller, D. P., Koeve, W., and Oschlies, A.: Could artificial ocean alkalinization
protect tropical coral ecosystems from ocean acidification?, Environ Res Lett, 11, Artn
074008, doi:10.1088/1748-9326/11/7/074008, 2016.
Frolicher, T. L., and Joos, F.: Reversible and irreversible impacts of greenhouse gas
emissions in multi-century projections with the NCAR global coupled carbon cycle-climate
model, Clim Dynam, 35, 1439-1459, Doi 10.1007/S00382-009-0727-0, 2010.
Fuss, S., Canadell, J. G., Peters, G. P., Tavoni, M., Andrew, R. M., Ciais, P., Jackson, R. B.,
Jones, C. D., Kraxner, F., Nakicenovic, N., Le Quere, C., Raupach, M. R., Sharifi, A., Smith,





P., and Yamagata, Y.: Commentary: Betting on Negative Emissions, Nat Clim Change, 4, 850-853, 2014.

Gasser, T., Guivarch, C., Tachiiri, K., Jones, C. D., and Ciais, P.: Negative emissions physically needed to keep global warming below 2 degrees C, Nat Commun, 6, Artn 7958 10.1038/Ncomms8958, 2015.

Gattuso, J.-P., Magnan, A., Billé, R., Cheung, W. W. L., Howes, E. L., Joos, F., Allemand, D., Bopp, L., Cooley, S. R., Eakin, C. M., Hoegh-Guldberg, O., Kelly, R. P., Pörtner, H.-O., Rogers, A. D., Baxter, J. M., Laffoley, D., Osborn, D., Rankovic, A., Rochette, J., Sumaila, U. R., Treyer, S., and Turley, C.: Contrasting futures for ocean and society from different anthropogenic CO2 emissions scenarios, Science, 349, 10.1126/science.aac4722, 2015.

Gonzalez, M. F., and Ilyina, T.: Impacts of artificial ocean alkalinization on the carbon cycle and climate in Earth system simulations, Geophys Res Lett, 43, 6493-6502, 10.1002/2016GL068576, 2016.

Groeskamp, S., Lenton, A., Matear, R., Sloyan, B. M., and Langlais, C.: Anthropogenic carbon in the oceanSurface to interior connections, Global Biogeochemical Cycles, 30, 1682-1698, 10.1002/2016GB005476, 2016.

Hauck, J., Kohler, P., Wolf-Gladrow, D., and Volker, C.: Iron fertilisation and century-scale effects of open ocean dissolution of olivine in a simulated CO2 removal experiment, Environ Res Lett, 11, Artn 024007, doi:10.1088/1748-9326/11/2/024007, 2016.

Heinze, C.: Simulating oceanic CaCO3 export production in the greenhouse, Geophys. Res. Let., 31, 2004.

Hoegh-Guldberg, O., Mumby, P. J., Hooten, A. J., Steneck, R. S., Greenfield, P., Gomez, E., Harvell, C. D., Sale, P. F., Edwards, A. J., Caldeira, K., Knowlton, N., Eakin, C. M., Iglesias-Prieto, R., Muthiga, N., Bradbury, R. H., Dubi, A., and Hatziolos, M. E.: Coral reefs under rapid climate change and ocean acidification, Science, 318, 1737-1742, 10.1126/science.1152509, 2007.

Hughes, T. P., Kerry, J. T., Alvarez-Noriega, M., Alvarez-Romero, J. G., Anderson, K. D., Baird, A. H., Babcock, R. C., Beger, M., Bellwood, D. R., Berkelmans, R., Bridge, T. C., Butler, I. R., Byrne, M., Cantin, N. E., Comeau, S., Connolly, S. R., Cumming, G. S., Dalton, S. J., Diaz-Pulido, G., Eakin, C. M., Figueira, W. F., Gilmour, J. P., Harrison, H. B., Heron, S. F., Hoey, A. S., Hobbs, J. P. A., Hoogenboom, M. O., Kennedy, E. V., Kuo, C. Y., Lough, J. M., Lowe, R. J., Liu, G., Cculloch, M. T. M., Malcolm, H. A., Mcwilliam, M. J., Pandolfi, J. M., Pears, R. J., Pratchett, M. S., Schoepf, V., Simpson, T., Skirving, W. J., Sommer, B., Torda, G., Wachenfeld, D. R., Willis, B. L., and Wilson, S. K.: Global warming and recurrent mass bleaching of corals, Nature, 543, 373-+, 10.1038/nature21707, 2017.

Iglesias-Rodriguez, M. D., Halloran, P. R., Rickaby, R. E. M., Hall, I. R., Colmenero-Hidalgo, E., Gittins, J. R., Green, D. R. H., Tyrrell, T., Gibbs, S. J., von Dassow, P., Rehm, E., Armbrust, E. V., and Boessenkool, K. P.: Phytoplankton calcification in a high-CO2 world, Science, 320, 336-340, Doi 10.1126/Science.1154122, 2008.

Ilyina, T., Wolf-Gladrow, D., Munhoven, G., and Heinze, C.: Assessing the potential of calcium-based artificial ocean alkalinization to mitigate rising atmospheric CO2 and ocean acidification, Geophys Res Lett, 40, 5909-5914, 10.1002/2013GL057981, 2013.

Jickells, T. D., An, Z. S., Andersen, K. K., Baker, A. R., Bergametti, G., Brooks, N., Cao, J. J., Boyd, P. W., Duce, R. A., Hunter, K. A., Kawahata, H., Kubilay, N., laRoche, J., Liss, P. S., Mahowald, N., Prospero, J. M., Ridgwell, A. J., Tegen, I., and Torres, R.: Global iron connections between desert dust, ocean biogeochemistry, and climate, Science, 308, 67-71, Doi 10.1126/Science.1105959, 2005.

Jones, C. D., Arora, V., Friedlingstein, P., Bopp, L., Brovkin, V., Dunne, J., Graven, H., Hoffman, F., Ilyina, T., John, J. G., Jung, M., Kawamiya, M., Koven, C., Pongratz, J., Raddatz, T., Randerson, J. T., and Zaehle, S.: C4MIP-The Coupled Climate-Carbon Cycle




Model Intercomparison Project: experimental protocol for CMIP6, Geosci Model Dev, 9,
2853-2880, 10.5194/gmd-9-2853-2016, 2016.
Keller, D. P., Feng, E. Y., and Oschlies, A.: Potential climate engineering effectiveness and
side effects during a high carbon dioxide-emission scenario, Nat Commun, 5, Artn 3304
10.1038/Ncomms4304, 2014.
Keller, D. P., Lenton, A., Scott, V., Vaughan, N. E., Bauer, N., Ji, D., Jones, C., Kravitz, B.,
Muri, H., and Zickfeld, K.: The Carbon Dioxide Removal Model Intercomparison Project
(CDR-MIP): Rationale and experimental design, Geoscientific Model Development
Discussions, 2017.
Kheshgi, H. S.: Sequestering Atmospheric Carbon-Dioxide by Increasing Ocean Alkalinity,
Energy, 20, 915-922, Doi 10.1016/0360-5442(95)00035-F, 1995.
Kohler, P., Abrams, J. F., Volker, C., Hauck, J., and Wolf-Gladrow, D. A.: Geoengineering
impact of open ocean dissolution of olivine on atmospheric $CO_2$, surface ocean pH and
marine biology, Environ Res Lett, 8, doi:10.1088/1748-9326/8/1/014009, 2013.
Le Quéré, C., Moriarty, R., Andrew, R. M., Peters, G. P., Ciais, P., Friedlingstein, P., Jones,
S. D., Sitch, S., Tans, P., Arneth, A., Boden, T. A., Bopp, L., Bozec, Y., Canadell, J. G.,
Chini, L. P., Chevallier, F., Cosca, C. E., Harris, I., Hoppema, M., Houghton, R. A., House, J.
I., Jain, A. K., Johannessen, T., Kato, E., Keeling, R. F., Kitidis, V., Klein Goldewijk, K.,
Koven, C., Landa, C. S., Landschützer, P., Lenton, A., Lima, I. D., Marland, G., Mathis, J.
T., Metzl, N., Nojiri, Y., Olsen, A., Ono, T., Peng, S., Peters, W., Pfeil, B., Poulter, B.,
Raupach, M. R., Regnier, P., Rödenbeck, C., Saito, S., Salisbury, J. E., Schuster, U.,
Schwinger, J., Séférian, R., Segschneider, J., Steinhoff, T., Stocker, B. D., Sutton, A. J.,
Takahashi, T., Tilbrook, B., van der Werf, G. R., Viovy, N., Wang, Y. P., Wanninkhof, R.,
Wiltshire, A., and Zeng, N.: Global carbon budget 2014, 7, 47-85, 2015.
Lenton, A., and Matear, R. J.: The role of the Southern Annular Mode (SAM) in Southern
Ocean $CO_2$ uptake, Global Biogeochemical Cycles, 21, doi: 10:1029/2006GB002714, 2007.
Lenton, A., Tilbrook, B., Matear, R. J., Sasse, T. P., and Nojiri, Y.: Historical reconstruction
of ocean acidification in the Australian region, Biogeosciences, 13, 1753-1765, 10.5194/bg-
786    13-1753-2016, 2016.
Lenton, A., Keller, D. P., and Pfister, P.: How Will Earth Respond to Plans for Carbon
Dioxide Removal?, EOS, 98, 10.1029/2017EO068385, 2017.
Lovelock, J. E., and Rapley, C. G.: Ocean pipes could help the Earth to cure itself, Nature,
449, 403-403, 10.1038/449403a, 2007.
Mao, J., Phipps, S. J., Pitman, A. J., Wang, Y. P., Abramowitz, G., and Pak, B.: The CSIRO
Mk3L climate system model v1.0 coupled to the CABLE land surface scheme v1.4b:
evaluation of the control climatology, Geosci Model Dev, 4, 1115-1131, 10.5194/gmd-4-
794    1115-2011, 2011.
Matear, R. J., and Hirst, A. C.: Long term changes in dissolved oxygen concentrations in the
ocean caused by protracted global warming, Global Biogeochemical Cycles, 17, 1125, 2003.
Matear, R. J., and Lenton, A.: Quantifying the impact of ocean acidification on our future
climate, Biogeosciences, 11, 3965-3983, Doi 10.5194/Bg-11-3965-2014, 2014.
Mathesius, S., Hofmann, M., Caldeira, K., and Schellnhuber, H. J.: Long-term response of
oceans to $CO_2$ removal from the atmosphere, Nat Clim Change, 5, 1107-+,
10.1038/NCLIMATE2729, 2015.
Mongin, M., Baird, M. E., Hadley, S., and Lenton, A.: Optimising reef-scale $CO_2$ removal by
seaweed to buffer ocean acidification, Environ Res Lett, 11, Artn 034023
10.1088/1748-9326/11/3/034023, 2016a.
Mongin, M., Baird, M. E., Tilbrook, B., Matear, R. J., Lenton, A., Herzfeld, M., Wild-Allen,
K., Skerratt, J., Margvelashvili, N., Robson, B. J., Duarte, C. M., Gustafsson, M. S. M.,



Ralph, P. J., and Steven, A. D. L.: The exposure of the Great Barrier Reef to ocean
acidification, Nat Commun, 7, Artn 10732
10.1038/Ncomms10732, 2016b.
Montserrat, F., Renforth, P., Hartmann, J., Leermakers, M., Knops, P., and Meysman, F. J.
R.: Olivine Dissolution in Seawater: Implications for CO2 Sequestration through Enhanced
Weathering in Coastal Environments, Environmental science & technology, 51, 3960-3972,
10.1021/acs.est.605942, 2017.
Mucci, A.: The Solubility of Calcite and Aragonite in Seawater at Various Salinities,
Temperatures, and One Atmosphere Total Pressure, Am J Sci, 283, 780-799, 1983.
Munday, P. L., Donelson, J. M., Dixson, D. L., and Endo, G. G. K.: Effects of ocean
acidification on the early life history of a tropical marine fish, P Roy Soc B-Biol Sci, 276,
3275-3283, Doi 10.1098/Rspb.2009.0784, 2009.
Munday, P. L., Dixson, D. L., McCormick, M. I., Meekan, M., Ferrari, M. C. O., and
Chivers, D. P.: Replenishment of fish populations is threatened by ocean acidification, P Natl
Acad Sci USA, 107, 12930-12934, Doi 10.1073/Pnas.1004519107, 2010.
National Research Council: Climate Intervention: Carbon Dioxide and the Reliable
Sequestration of Carbon Washington D.C., US, 2015.
Orr, J. C., Fabry, V. J., Aumont, O., Bopp, L., Doney, S. C., Feely, R. A., Gnanadesikan, A.,
Gruber, N., Ishida, A., Joos, F., Key, R. M., Lindsay, K., Maier-Reimer, E., Matear, R.,
Monfray, P., Mouchet, A., Najjar, R. G., Plattner, G. K., Rodgers, K. B., Sabine, C. L.,
Sarmiento, J. L., Schlitzer, R., Slater, R. D., Totterdell, I. J., Weirig, M. F., Yamanaka, Y.,
and Yool, A.: Anthropogenic ocean acidification over the twenty-first century and its impact
on calcifying organisms, Nature, 437, 681-686, 2005.
Phipps, S. J., Rotstayn, L. D., Gordon, H. B., Roberts, J. L., Hirst, A. C., and Budd, W. F.:
The CSIRO Mk3L climate system model version 1.0-Part 2: Response to external forcings,
Geosci Model Dev, 5, 649-682, 10.5194/gmd-5-649-2012, 2012.
Ragueneau, O., Treguer, P., Leynaert, A., Anderson, R. F., Brzezinski, M. A., DeMaster, D.
J., Dugdale, R. C., Dymond, J., Fischer, G., Francois, R., Heinze, C., Maier-Reimer, E.,
Martin-Jezequel, V., Nelson, D. M., and Queguiner, B.: A review of the Si cycle in the
modem ocean: recent progress and missing gaps in the application of biogenic opal as a
paleoproductivity proxy, Global Planet Change, 26, 317-365, Doi 10.1016/S0921-
838  8181(00)00052-7, 2000.
Raven, J., Caldeira, K., Elderfield, H., Hoegh-Guldberg, O., Liss, P., and Riebesell, U.:
Ocean acidification due to increasing atmospheric carbon dioxide. The Royal Society, Policy
Document, London, UK, 2005.
Renforth, P., and Henderson, G.: Assessing ocean alkalinity for carbon sequestration, Review
in Geophysics, 55, 10.1002/2016RG000533, 2017.
Revelle, R., and Suess, H. E.: Carbon dioxide exchange between atmosphere and ocean and
the question of an increase of atmospheric CO2 during the past decades, Tellus, 9, 18-27,
846  1957.
Riebesell, U., Zondervan, I., Rost, B., Tortell, P. D., Zeebe, R. E., and Morel, F. M. M.:
Reduced calcification of marine plankton in response to increased atmospheric CO2, Nature,
849  407, 364-367, 2000.
Riebesell, U., Fabry, V. J., Hansson, L., and Gattuso, J.-P.: Guide to best practices for ocean
acidification research and data reporting, Publications Office of the European Union.,
Luxembourg, 260, 2010.
Rogelj, J., Meinshausen, M., and Knutti, R.: Global warming under old an new scenarios
using IPCC climate sensitivity range estimates, Nat Clim Change, 2, 248-253,
10.1038/NCLIMATE1385, 2012.



Rogelj, J., den Elzen, M., Hohne, N., Fransen, T., Fekete, H., Winkler, H., Chaeffer, R. S.,
Ha, F., Riahi, K., and Meinshausen, M.: Paris Agreement climate proposals need a boost to
keep warming well below 2 degrees C, Nature, 534, 631-639, 10.1038/nature18307, 2016.
Sabine, C. L., Feely, R. A., Gruber, N., Bullister, J. L., Wanninkhof, R., Wong, C. S.,
Wallace, D. W. R., Tilbrook, B., Millero, F. J., Peng, T.-H., Kozyr, A., Ono, T., and Rios, A.
F.: The Oceanic Sink for Anthropogenic $CO_2$, Science, 305, 367-371, 2004.
Scott, V., Haszeldine, R. S., Tett, S. F. B., and Oschlies, A.: Fossil fuels in a trillion tonne
world, Nat Clim Change, 5, 419-423, 10.1038/NCLIMATE2578, 2015.
Seneviratne, S. I., Corti, T., Davin, E. L., Hirschi, M., Jaeger, E. B., Lehner, I., Orlowsky, B.,
and Teuling, A. J.: Investigating soil moisture-climate interactions in a changing climate: A
review, Earth-Science Reviews, 99, 125-161, 10.1016/j.earscirev.2010.02.004, 2010.
Sigman, D. M., and Boyle, E. A.: Glacial/interglacial variations in atmospheric carbon
dioxide, Nature, 407, 859-869, Doi 10.1038/35038000, 2000.
Smith, P., Davis, S. J., Creutzig, F., Fuss, S., Minx, J., Gabrielle, B., Kato, E., Jackson, R. B.,
Cowie, A., Kriegler, E., van Vuuren, D. P., Rogelj, J., Ciais, P., Milne, J., Canadell, J. G.,
McCollum, D., Peters, G., Andrew, R., Krey, V., Shrestha, G., Friedlingstein, P., Gasser, T.,
Grubler, A., Heidug, W. K., Jonas, M., Jones, C. D., Kraxner, F., Littleton, E., Lowe, J.,
Moreira, J. R., Nakicenovic, N., Obersteiner, M., Patwardhan, A., Rogner, M., Rubin, E.,
Sharifi, A., Torvanger, A., Yamagata, Y., Edmonds, J., and Cho, Y.: Biophysical and
economic limits to negative $CO2$ emissions, Nat Clim Change, 6, 42-50,
10.1038/NCLIMATE2870, 2016.
Society, T. R.: Geoengineering the Climate System: Sceonce Goverence and Uncertainity,
London, UK, 2009.
Taylor, K. E., Stouffer, R. J., and Meehl, G. A.: An Overview of Cmip5 and the Experiment
Design, B Am Meteorol Soc, 93, 485-498, 10.1175/BAMS-D-11-00094.1, 2012.
Trull, T. W., Rintoul, S. R., Hadfield, M., and Abraham, E. R.: Circulation and seasonal
evolution of polar waters south of Australia: Implications for iron fertilizations for the
Southern Ocean, Deep-Sea Research II, 48, 2439-2466, 2001.
United Nations Framework on Climate Change: Adoption of the Paris Agreement, 21st
Conference of the Parties. 2015.
Wang, Y. P., Law, R. M., and Pak, B.: A global model of carbon, nitrogen and phosphorus
cycles for the terrestrial biosphere, Biogeosciences, 7, 2261-2282, 10.5194/bg-7-2261-2010,
888  2010.
Yamamoto, A., Kawamiya, M., Ishida, A., Yamanaka, Y., and Watanabe, S.: Impact of rapid
sea-ice reduction in the Arctic Ocean on the rate of ocean acidification, Biogeosciences, 9,
2365-2375, 10.5194/bg-9-2365-2012, 2012.
Yamanaka, Y., and Tajika, E.: The role of vertical fluxes of particulate organic material and
calcite in the oceanic carbon cycle: Studies using a ocean biogeochemical general; circulation
model, Global Biogeochemical Cycles, 10, 361-382, 1996.
Yool, A., Popova, E. E., and Coward, A. C.: Future change in ocean productivity: Is the
Arctic the new Atlantic?, J Geophys Res-Oceans, 120, 7771-7790, 10.1002/2015JC011167,
897  2015.
Zeebe, R. E.: History of Seawater Carbonate Chemistry, Atmospheric $CO2$, and Ocean
Acidification, Annu Rev Earth Pl Sc, 40, 141-165, 10.1146/annurev-earth-042711-105521,
900  2012.
Zhang, Q., Wang, Y. P., Matear, R. J., Pitman, A. J., and Dai, Y. J.: Nitrogen and
phosphorous limitations significantly reduce future allowable $CO2$ emissions, Geophys Res
Lett, 41, 632-637, 10.1002/2013GL058352, 2014a.
Zhang, Z. G., Wang, W., and Qiu, B.: Oceanic mass transport by mesoscale eddies, Science,
345, 322-324, 10.1126/science.1252418, 2014b.



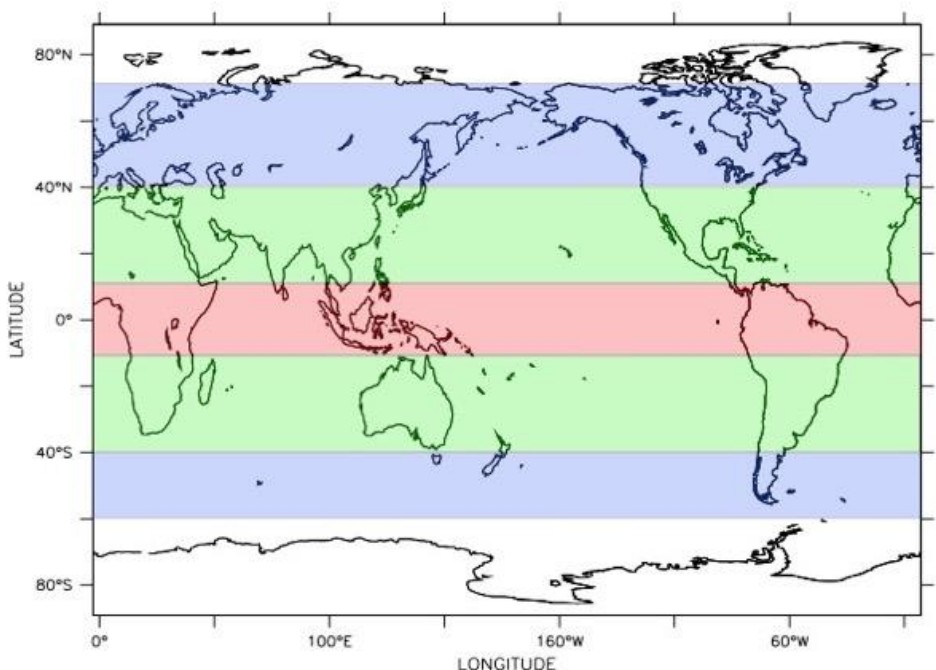


Figure 1 Ocean regions used for Alkalinity Injection in the period 2020-2100, the blue
denotes the subpolar regions (AOA_SP), the green regions represent the subtropical gyres
(AOA_ST), red the tropical ocean (AOA_T), and all colored regions combined the global
alkalinity injection (AOA_G). Note that the ocean regions not colored represent the seasonal
sea-ice, where no alkalinity was added in the simulation.



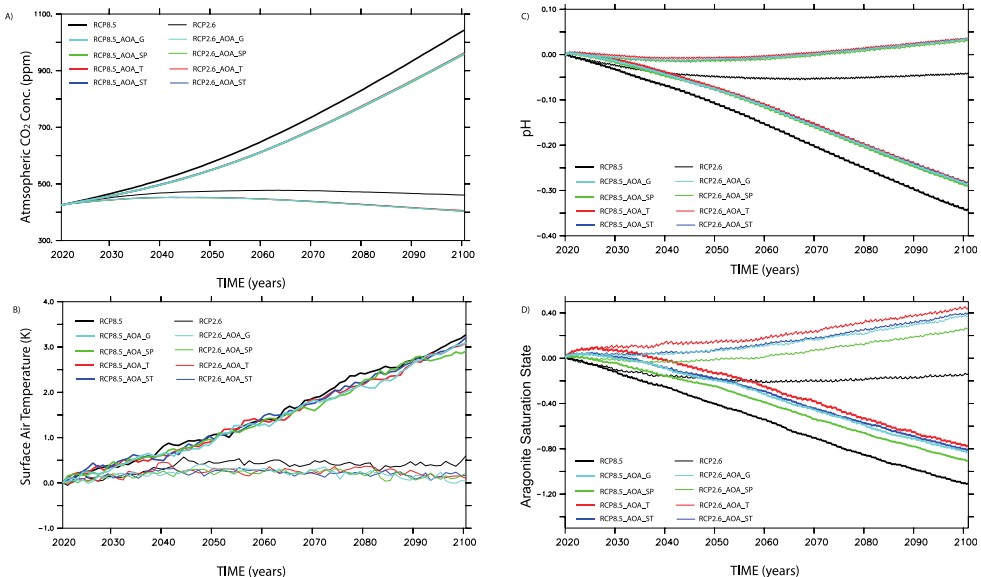



Figure 2 The global mean changes in: Atmospheric CO2 concentration (A), Surface Air
Temperature (SAT; B), surface ocean pH (C) and Aragonite Saturation State (D) for high
(RCP8.5) and low emissions (RCP2.6) with global and regional AOA in the period 2020-

919    2100.




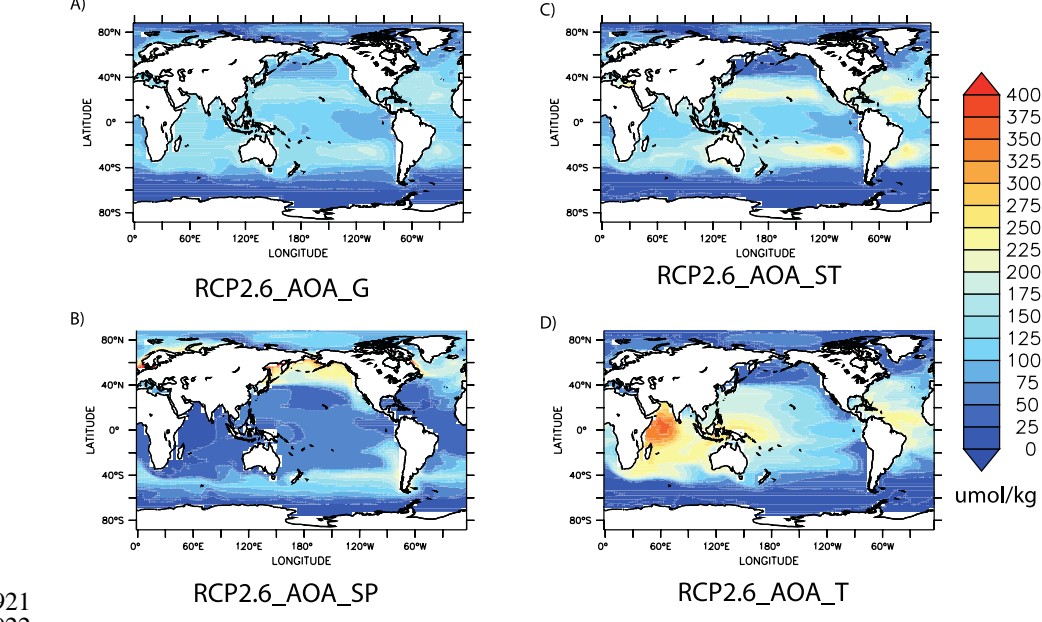



Figure 3 The spatial map of relative increase in surface alkalinity in 2090 (mean; 2081-2100)
associated with global and regional AOA under RCP2.6. Units are µmol/kg

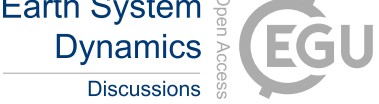



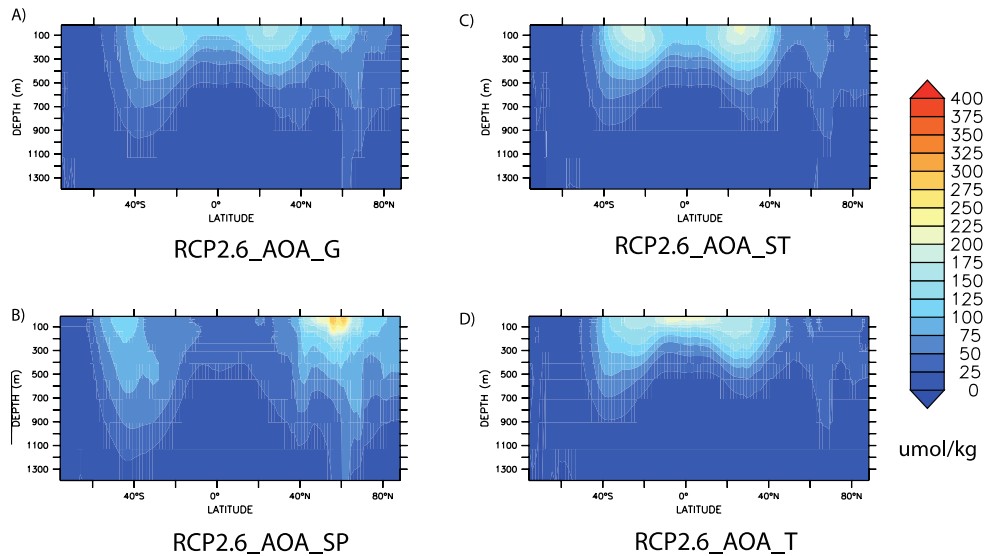

Figure 4 The zonal mean relative changes in alkalinity in the interior ocean associated with
global and regional AOA under RCP8.5 in 2090 (mean:2081-2100). Units are μmol/kg.





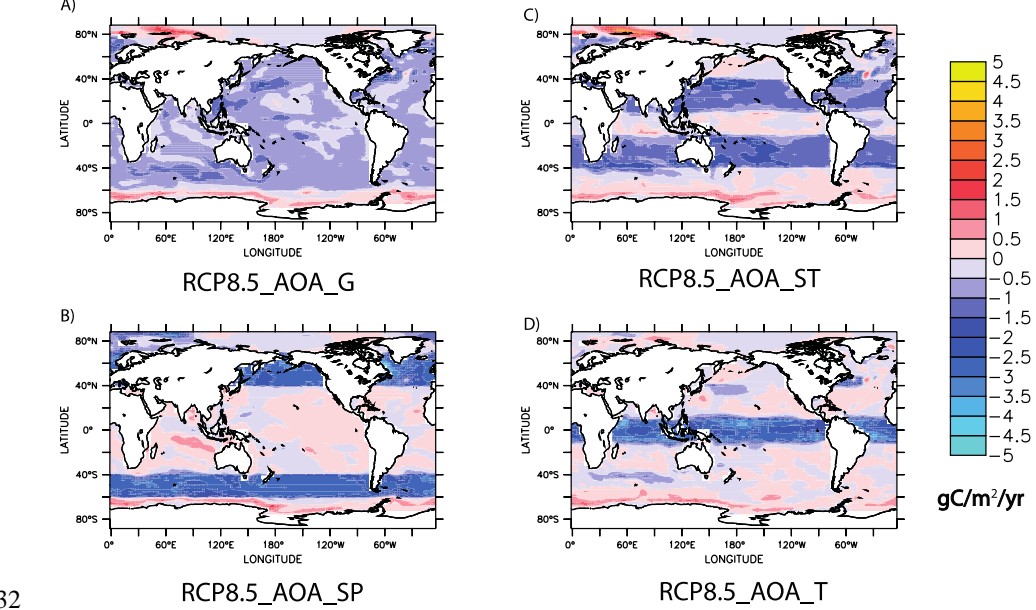



Figure 5 The spatial map of relative changes in ocean carbon uptake in 2090 (mean; 2081-
2100) associated with global and regional AOA under RCP8.5. Units are gC/m²/yr






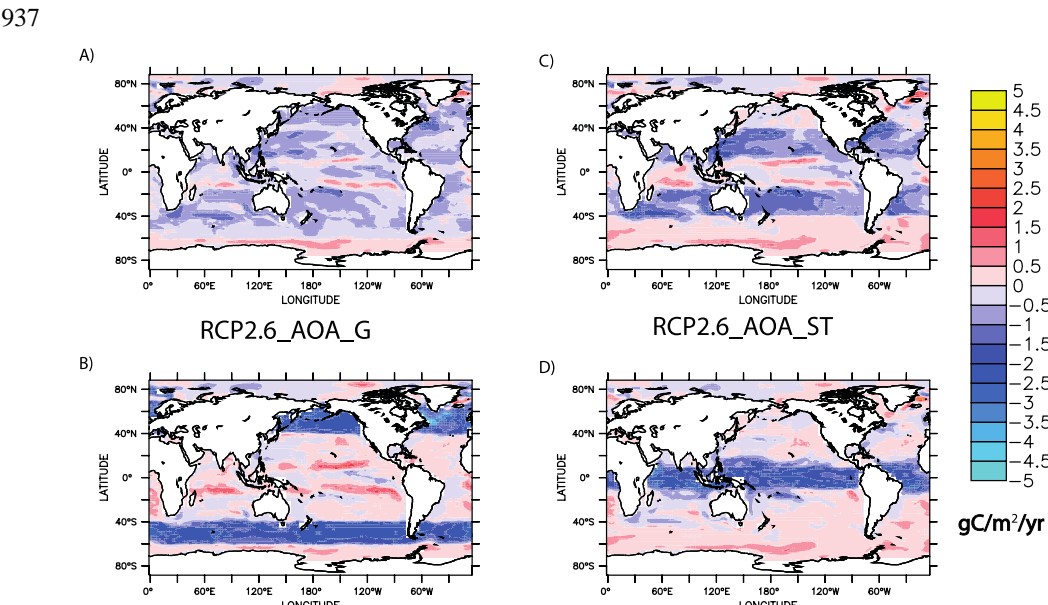


Figure 6 The spatial map of relative changes in ocean carbon uptake in 2090 (mean; 2081-
2100) associated with global and regional AOA under RCP2.6. Units are gC/m²/yr




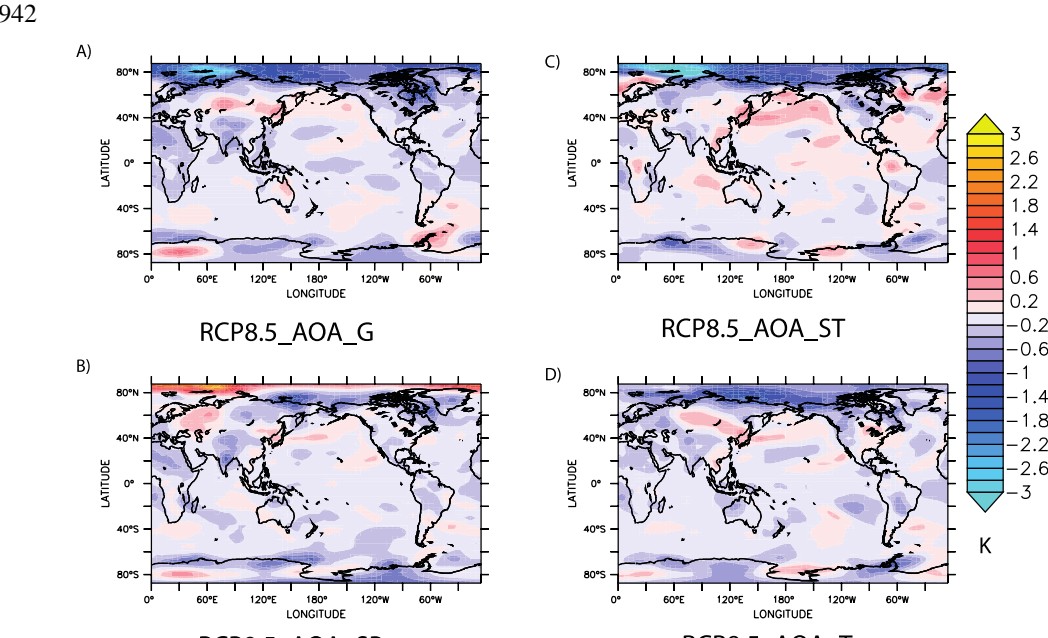


Figure 7 The spatial map of relative changes in surface air temperature 2090 (mean; 2081-
2100) associated with global and regional AOA under RCP8.5. Units are K



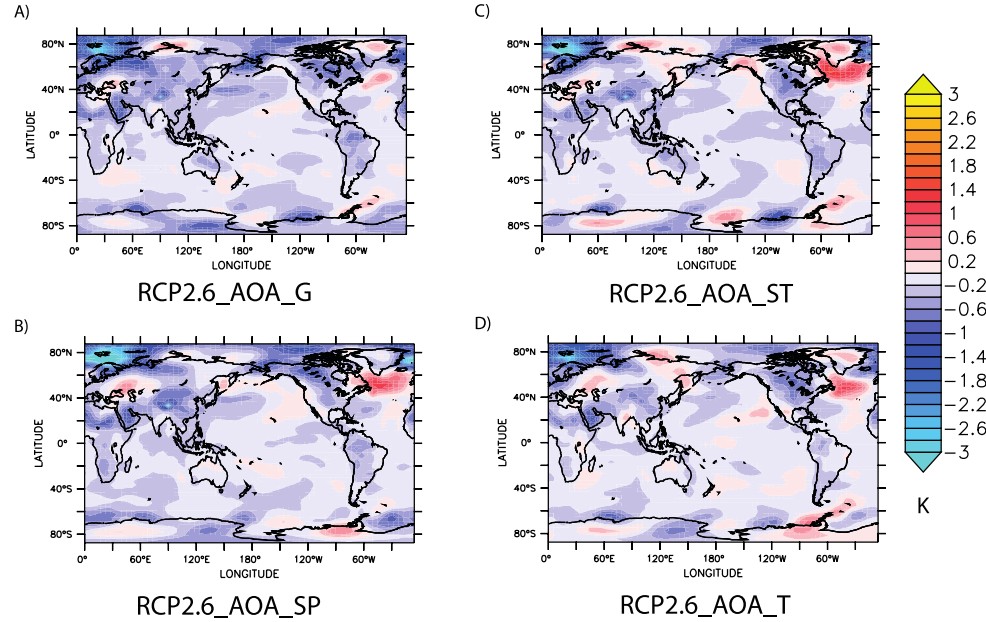



Figure 8 The spatial map of relative changes in surface air temperature 2090 (mean; 2081-
2100) associated with global and regional AOA under RCP2.6. Units are K





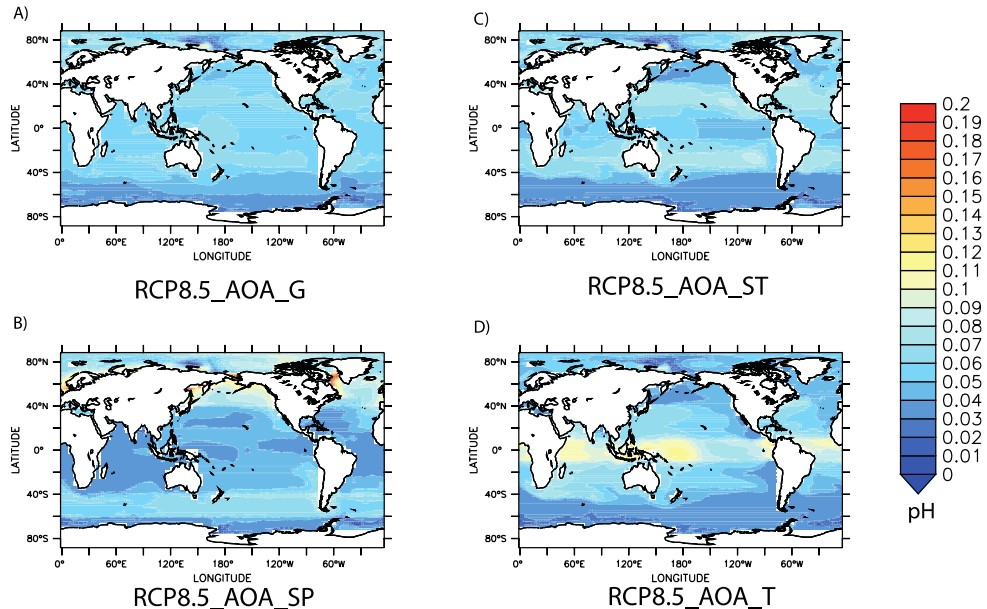



Figure 9 The spatial map of the relative changes in pH in 2090 (mean; 2081-2100) associated
with global and regional AOA under RCP8.5.





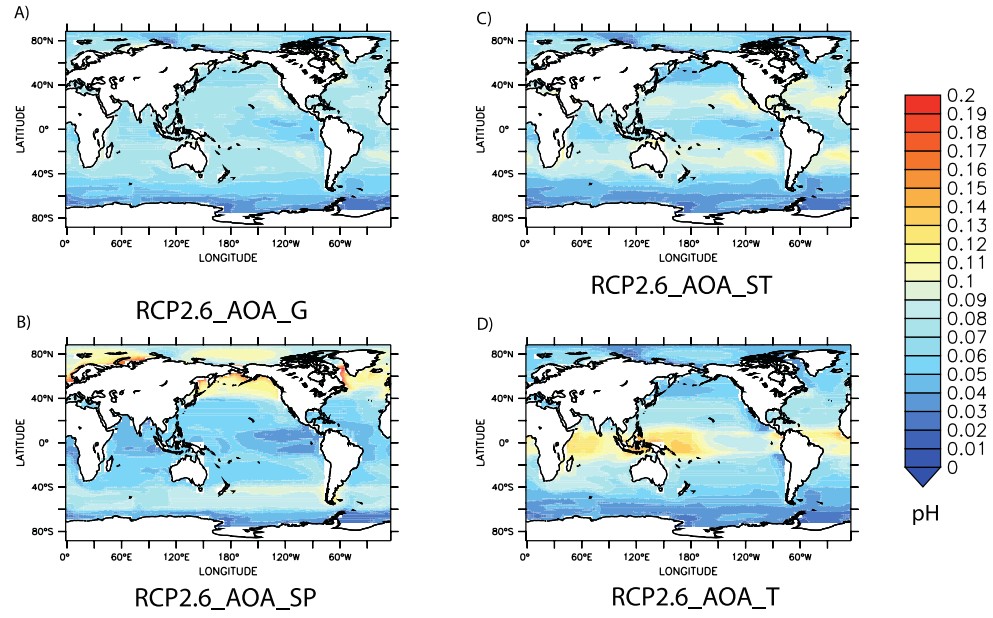



Figure 10 The spatial map of the relative changes in pH in 2090 (mean; 2081-2100)
associated with global and regional AOA under RCP2.6.





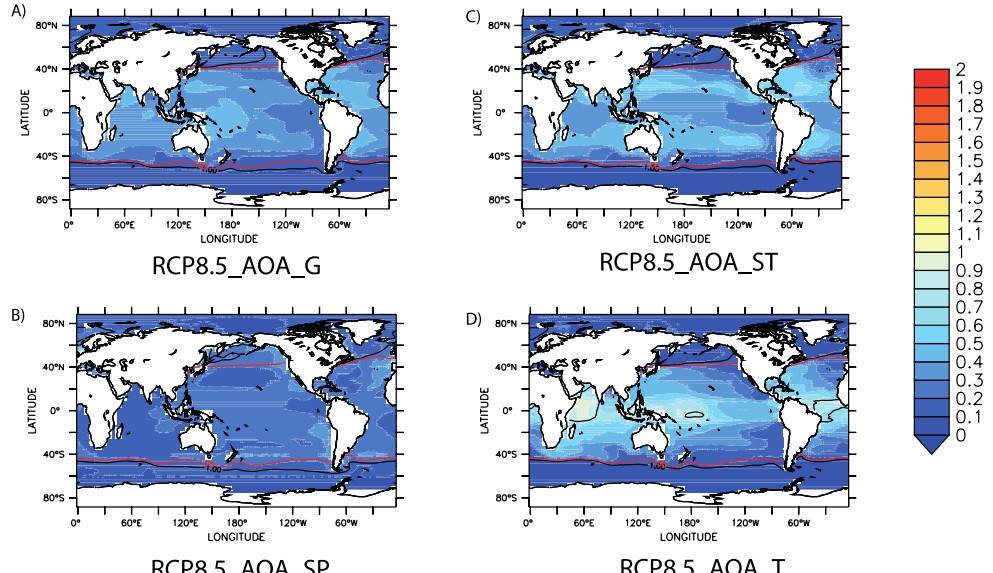



Figure 11 The spatial map of the relative differences in surface aragonite saturation state in
2090 (mean 2081-2100), associated with global and regional AOA under RCP8.5. Contoured
on each map are the values of aragonite saturation state of 1 and 3, please see the text for
more explanation. The red contours represent RCP8.5 and black AOA for each experiment



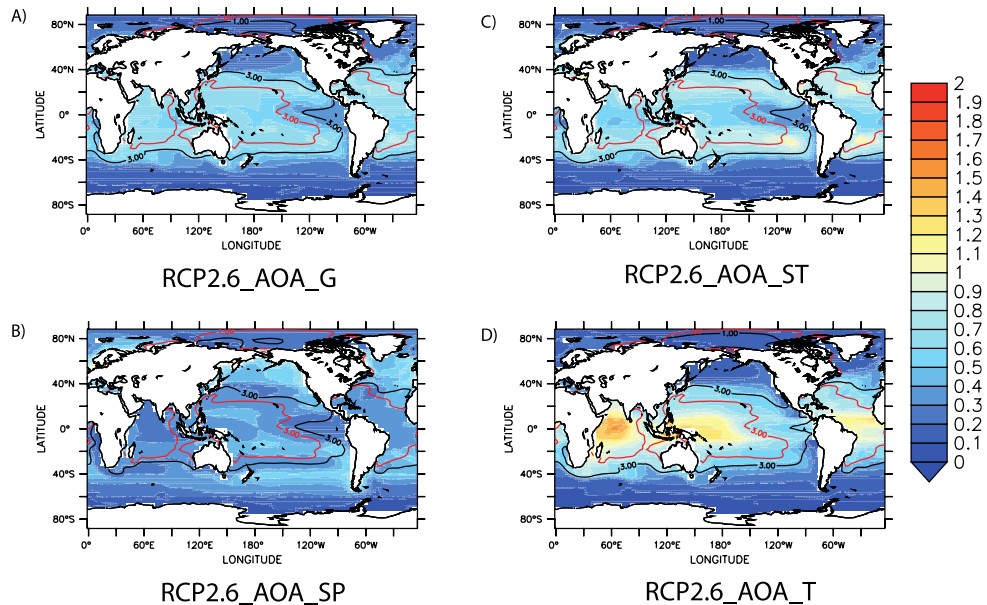



Figure 12 The spatial map of the relative differences in surface aragonite saturation state in

2090 (mean 2081-2100), associated with global and regional AOA under RCP2.6. Contoured

on each map are the values of aragonite saturation state of 1 and 3, please see the text for

more explanation. The red contours represent RCP2.6 and black AOA for each experiment

975



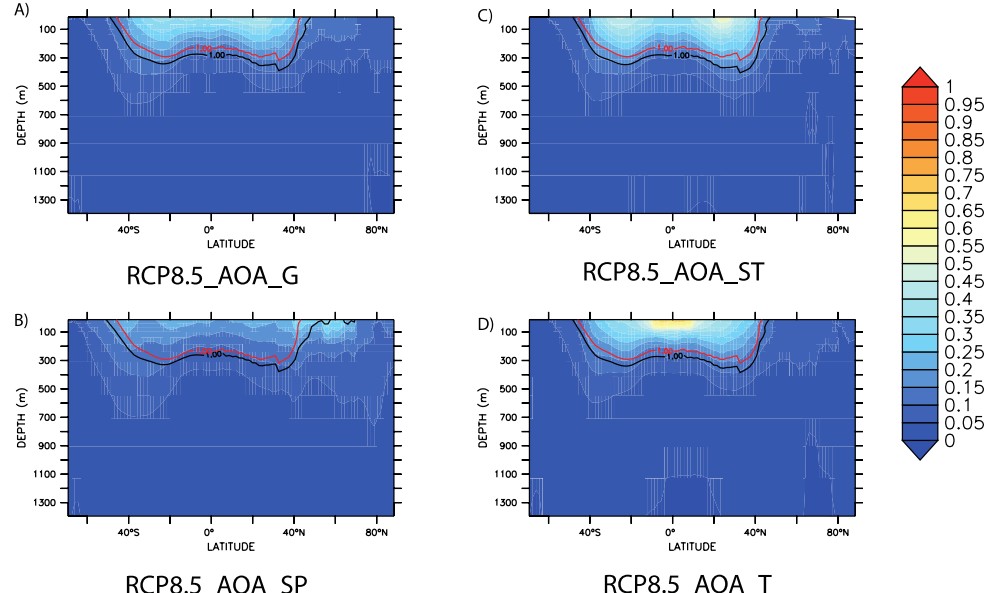

976

Figure 13 The relative zonal mean differences in aragonite saturation state in 2090 (mean

2081-2100), associated with global and regional AOA under RCP8.5. Contoured on each

map are the values of aragonite saturation state of 1, please see the text for more explanation.

The red contours represent RCP8.5 and black AOA for each experiment.






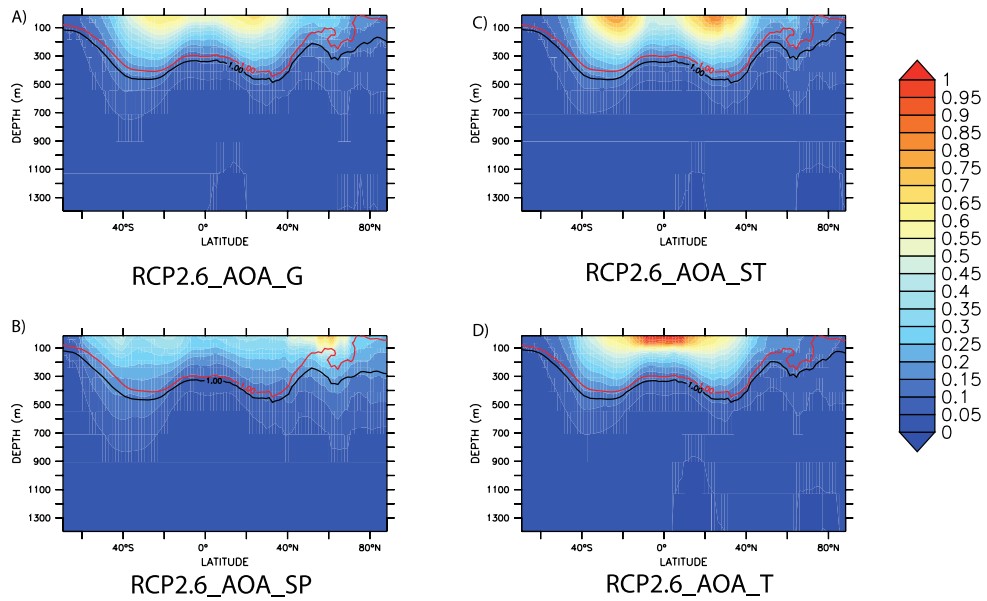



Figure 14 The relative zonal mean differences in aragonite saturation state in 2090 (mean
2081-2100), associated with global and regional AOA under RCP2.6. Contoured on each
map are the values of aragonite saturation state of 1, please see the text for more explanation.
The red contours represent RCP2.6 and black AOA for each experiment.