# Peer review of "Assessing Carbon Dioxide Removal Through Global and Regional Ocean Alkalization"

_Earth System Dynamics, 2017_

## Referee Comment (RC1) · Anonymous Referee #1 · 28 Nov 2017

I enjoyed reading this manuscript. The authors make a valuable contribution to an emerging field. Given that this is a relatively new area, this paper may set the normative approach to assessing and interpreting AOA model experiments. It would be useful if the authors took some time to explore the potential range of experiments (addition amounts, location of addition, against a wider range of emissions scenarios, overshoot/recover), what has already been addressed by previous work, and what is still left to do (potentially this could be tabulated?). Generally, this work is excellent and would encourage publication, with some minor edits/comments below.

Abstract: L19-21: Be specific, what changes are seen? In what parameter?

[Figure]

L21-22: Not quite sure what that means.

L22-23 The change in saturation state is ambiguously describe, refer specifically to changes in omega.

L28 It's left a little open ended here, you could be more specific with the regional response. It is one of the more important findings from the experiment.

Introduction: Good introduction. Clearly explains why we need CDR and more specifically AOA. Also gives a description of ocean acidification and how AOA works.

L50: 'Including through coral bleaching' a little clunky, maybe remove 'through'

L54: Could you say something about the changing Revelle Factor, and the potential for AOA to impact this?

L148-150: simply states "impact" which could be a bit vague. Could go in further and state that they will be investigating the impact on the "carbon cycle, global surface warming (2m surface air temperature), and response and ocean acidification response to the 4 different AOA experiments under the high (RCP8.5) and low (RCP2.6) emissions scenarios." Which is stated in lines 207-210. Regarding my comment above, it is worth exploring the potential experiment space, magnitude of alkalinity addition, location, emission scenario, and the resulting impacts site specific/regional/global/ open ocean/coastal etc. What parts of this picture does your model/this paper deal with, what has already been done by others, and what is left to do? are other models needed?

Methods:

Model seems appropriate for the scope of this paper. Clear description of the experimental design which seems appropriate to answer the research question proposed in the introduction. Could explain what the model outputs are? Also should mention the testing for seasonality? (mentioned in lines 538-554)

L162-164: Do you expect this assumption to hold up under elevated alkalinity? Could

the rain ratio change?

L204: Fair assumption, but it is worth pointing out that alkalinity manipulation could be from carbonate dissolution or NaOH addition which would not induce and impact from iron and silicate. You are testing the fundamental impact intrinsic to all of these methods of C sequestration. Results and discussion:

L208 – 209: the sentence doesn't make sense, a typo somewhere?

L215: Why have you chose this addition rate? Also, should alkalinity not be expressed in equivalents rather than moles (and throughout)?

L218-221: Fascinating, but why was the response different?

L232: "at" is missing

L239: I think 'an overall' is missing before 525 ppm in the brackets

L241: could you also give this as a % similar to how you did for RCP8.5

L251-254: This is really important...why was there an increase in export?... The 1% is an important outcome because it is the 'efficiency reduction' on the overall engineering system design.

Could tables 1-3 be summarised in one table? I think it would make things a little clearer.

Line 331: could be explained more clearly rather than just "due to the Revelle factor" (see previous comment)

L374: 'This reflects the fact that' should be rewritten 'This is caused by the subduction processes..' or something similar.

L396: 'Quite low', how low?

L445-447: But could you speculate as you have in the previous sentence? How much would export have to change to make a material difference?

L507: This doesn't quite ring with your abstract, which suggests that ocean acidification would be mitigated. Would it not partially ameliorate the impacts?

L599: 'Interestingly' is used a bit too often, it gets a bit jarring.

Figures:

All Figures are clear. Slightly too many for this type of manuscript. Could some be moved to the supporting information? Figures 11-12 are not referred to in the text.

―――――――――――――――――――

---

## Referee Comment (RC2) · Anonymous Referee #2 · 12 Dec 2017

Lenton and colleagues explored the impacts of different artificial ocean alkalinization (AOA) scenarios on the global carbon cycle, surface temperatures and seawater carbonate chemistry under two contrasting Representative Concentration Pathways (RCPs), namely the "high" (emissions) RCP8.5 and "low" RCP2.6. They used the CSIRO-Mk3L-COAL model in its emissions driven configuration. Each AOA experiment simulated either a global (i.e. 60S-70N) or regional (i.e. subpolar northern and southern oceans (40S-60S; 40N-70N), subtropical oceans (15-40N; 15-40S) and equatorial oceans (15N-15S)) annual addition of 0.25 Pmol of alkalinity into the surface ocean.

This publication aims at answering:

a) Does the region in which AOA is implemented play a role in the associated (global and regional) response?

b) Does the state of the climate (mainly driven by the underlying RCP) make any difference in such a response to AOA?

These questions are really interesting and definitely worth studying. In general, their modelling approach is suitable for the purpose. However, I think that the authors still need to work on this manuscript in order to improve the presentation of the results and (even more important) explain them properly. I would not suggest changes in the order of the (sub)sections, but I strongly recommend that the narrative is modified since they are grammatical errors, typos and the current text does not seem to me reader-friendly. I do not find this manuscript ready for publication in its present form. In the following I describe my (major and minor) comments, from which I based my opinion.

Major comments:

Major comment 1: The changes in the land carbon uptake (table 2) in the AOA simulations based on the RCP2.6 are around 4 times higher than those of the simulations based on the RCP8.5. This is an important aspects because the variations in these carbon fluxes determine the final state of the climate. That is why I think that these results should be discussed properly and the cause of this differential behaviour should be explained.

Major comment 2: The statements given between the line 285 and 289 are really confusing. On the one hand, it reads as the temperature change in the RCP8.5 experiment is higher than the one associated with the RCP2.6, which is not what I see in the numbers. And on the other hand, making reference to "potentially reflecting feedbacks" in order to explain this cooling signal does not help to understand the signal. Instead, it confuses the reader. Please explain properly how these feedbacks affect the results.

Major comment 3: The reduction in atmospheric CO2 concentrations by 2100 associ-

ated with the AOA scenarios under RCP8.5 emissions (app. 84 ppm) is higher than the one associated with the AOA scenarios conducted under the RCP2.6 (app. 40 ppm). Yet, the mitigated warming in the AOA simulations under RCP2.6 is higher than those conducted under the RCP8.5. This is one of the main findings of this publication, however, there is not any discussion/explanation of this result. Only stating what the model delivers is not enough, since it could be a model artefact, the signal might not be caused by AOA, etc. The RCP8.5 and 2.6 scenarios have atmospheres with quite different levels of CO2, which might lead to differences in the CO2-forcing response to changes in CO2 levels. Not only that, but also the RCP8.5 and 2.6 scenarios differ in the assumed land use and the sea ice extent by the end of this century. This might also cause changes in albedo and therefore in the cooling response due to changes in forcing.

Major comment 4: Between the lines 325 and 334 an explanation to the differential pH and aragonite saturation state responses between simulations is given. This explanation seems confusing and it refers to the other main finding of this publication. Because of this I think that it requires some supporting figures (which could be added into the supplementary information) and some extra work in order to clarify the message. I suggest to look at the buffer factors and the effects of AOA under the two different DIC/ALK regimes associated with the RCP8.5 and 2.6 scenarios. More information can be found in the paper by Egleston et.al. (2010) (http://onlinelibrary.wiley.com/doi/10.1029/2008GB003407/abstract).

Minor comments:

L16, L27 and L561: "is capable of" gives the impression that AOA has not real big limitations to be implemented which is not the case, please modify the wording

L18, L19: there are acronyms which the reader might have never seen in the abstract, please spell them out or remove

L25: "lower" and "higher" emissions than what? I think that you meant "low" and "high"

L26: our simulations show that AOA during the period ... ; in any case I do not think that this very last sentence in the abstract is needed

L46: ... could help to ...

L53-54: CO2 that enters the ocean does not react with seawater to reduce the carbonate ion concentration, please reconsider this statement and use correct grammar

L59: ...changes in calcification...

L60: are you sure that ocean acidification alters nutrient availability

L62: please change order of Munday's cited publications

L69: semicolon needed?

L77: weathering of minerals play a crucial role in modulating the state of the climate in geological timescales, please write an assertive statement

L91: reviewed

L92-95: way too long sentence, please simplify and split it

L98: Did Kohler used one or several models?

L110: ocean only without the hyphen

L110: and they showed

L111: high CO2 emissions

L114: also concluded that

L115, L118, L131, ...: impacts of

L124: from a high

L126: it would be required

L127: and it would come

L134: 78 ppm between brackets might look better

L134: a net atmospheric cooling

L139: "to be very large" - (very) large in what respect? please clarify

L141: currently assume (instead of "utilize")

L141 to L145: way too long sentence, please simplify and split it

L149: and surface warming

L149: questions

L147 to L152: I think that the novelty of this study could be better emphasise. In any case, this last paragraph is crucial and therefore it should be improved since it does not read well.

L158: extra dot after citations?

L160, L163, ...: please remove the brackets in those citations which are subjects of the sentences, this occurs several times in this manuscript

L164 to L166: does this sentences really add any relevant information? Such a feature of the model is basic to conduct this study

L171 to L172: the land carbon cycle currently has too many uncertainties to state something in such an assertive manner, please consider to modify this or even remove it

L185: from 2006 onwards, ...

L186: corresponding to the Representative ...

L218: Subpolar addition

L232: is seen in 2100

L233 to L235: if you mention this feature of the modelling tool, please explain the associated consequences for the simulations of AOA

L242 and L256: "more than compensates" and "more than offset" are really confusing ways of describing the obtained values, please clarify

L247: 50% instead of 1.5 maybe?

L253: total ocean uptake ...

L254 to L258: this explanation reads really confusing, please clarify

L266: addition studies such as Ilyina ... which demonstrated ...

L270: 181 PgC is in ... (instead of was)

L277: I think the authors meant "positive denotes enhanced uptake" (instead of "negative")

L288: "large" twice in the sentence

L294: "projected" instead of "anticipated"

L295: why is this publication here cited?

L298: standard deviations with respect to what? what is this (1 - sigma)? Please clarify

L302 to L304: please consider to reformulate these sentences since "variability" might refer to many different things (e.g. inter annual, inter model, model internal, ...). In any case I think that "variability" is not really the term to use since what is described here are differences between simulations.

L308: mean surface cooling

L318: What is the point of this statement and citation? The pH and aragonite saturation state correlate really well as I can see in the figures.

L321: despite the return

L344 and L377: the citation here to Groeskamp et.al. seems unfounded

L348 and L349: Please elaborate on this so that the reader understand the context, e.g. discuss how this change in pH might (or not) matter, ...

L380: How can one of the experiments (AOA_ST) reflect the timescales of the circulation of the subtropical gyres? Please explain this.

L382: ice covered (instead of "non-ice-free")

L386: by 2100 (instead of "in")

L387 and L388: please clarify this, is not understandable

L404: ...seen in the...

L421 to L425: please work on the grammar of these sentences

L442: ... in the ratio ...

L444: Dot missing

L445: ... remain poorly...

L457 to L459: why do you obtain this result?

L463: remove (SAT)

L468 to L470: why do you obtain this result?

L538 to L554: why no figures are shown in this section on seasonality to support this discussion? Also, only AOA is implemented in the summer season under RCP8.5 emissions, which does not seem to me enough to explore the effects of seasonality.

L560: please remove (COP21)

L593 to L595: What do you mean? Please clarify this.

L605: double "that"

L620: mention "preindustrial period" and remove (1850) reads better

L621: cases (subject?) leads ...

L633: for the role

L638: ...therefore it needs ...

L642 and L645: Earth system (instead of earth system)

L649: please put "e.g. mesocosm experiments" between brackets

Please keep an eye on the format in which the references are given and be consistent with it.

---

## Referee Comment (RC3) · Anonymous Referee #3 · 21 Dec 2017

The Lenton et al., study investigates the impacts from adding artificial alkalinity to the oceans using the model CSIRO under 2 different emission pathways - RCP2.6 and RCP8.5. It was a really well done and interesting study to read technically, however my main comment is that the way the paper is currently structured makes it confusing to read. For example, each paragraph jumps back and forth between RCP26 and RCP85 making the story line hard to follow. I suggest setting up the story for one of the emissions pathways and then comparing to that one for the other pathway. It would also be useful to set up the chemistry in a little more detail or reiterate the paragraph in the intro. This would be useful when explaining why adding alk under a 2.6 scenario is more effective.

Lastly, section 3.1 was confusing (you may want to expand on the methods section to make this section clearer). For each run you added 0.25Pmol/yr of alkalinity but then I read in ln216 that the magnitude of the increase in alkalinity is dependent on where it was added. Is the 0.25Pmol/yr added to all the boxes? or is it divided up between the boxes for a total of 0.25Pmol/yr? Can you put everything in the same units to be constant?

Minor comments: ln50: "including through coral bleaching" - not clear what this means

ln79-80: This sentence seems out of place.

ln149: what do you mean by impact?

ln158: extra period between feedbacks and references

ln230: the first sentence does not make sense.

ln250: why is there a difference in export?

Section 3.1.2: I don't understand how soil-moisture feedbacks are influencing temperature in this case. Temperature is more variable on land than over the ocean, could it be simply that?

Table4: relative to what?

Figures: relative to what?

Figures: There are a lot of figures which I'm not sure add much to main text. You could simply the story in the text and only look at a few and toss the rest into the supplement. Or could you combine the 2.6 and 8.5 figures into 1? Could you do a difference between them? That would visually show the reader where the differences between scenarios lie....

Section 3.2.5: I think the figures referenced are not the correct figures. Plus I think a description of figures 11 and 12 are missing.

I see that the previous reviewers picked up a bunch of typos, missing words, and grammar mistakes so I won't repeat them here.

---

## Author Comment (AC1) · 24 Jan 2018

Response to Reviewer 1's comments for Assessing Carbon Dioxide Removal Through Global and Regional Ocean Alkalization under High and Low Emission Pathways by Lenton et al

Comments Reviewer 1 Minor Comments

Abstract: L19-21: Be specific, what changes are seen? In what parameter? Added "in alkalinity"

L21-22: Not quite sure what that means.

The sentence now reads: Globally, while we see that under RCP2.6 the carbon uptake associated with AOA is only ∼60% of the total under RCP8.5, the relative changes in temperature are larger, as are the changes in pH (140%) and aragonite saturation state (170%).

L22-23 The change in saturation state is ambiguously describe, refer specifically to changes in omega.

With respect the actual values are listed in the body of paper, and to list all of the values here would make the abstract too long.

L28 It's left a little open ended here, you could be more specific with the regional response. It is one of the more important findings from the experiment.

We would love to but given the length of the abstract we feel that we are somewhat limited in terms of space. But we have tried to be a little clearer, the last sentence of the abstract now states: Finally, our simulated AOA for 2020-2100 in the RCP2.6 scenario is capable of offsetting warming and ameliorating ocean acidification increases at the global scale, but with highly variable regional responses.

Introduction: Good introduction. Clearly explains why we need CDR and more specifically AOA. Also gives a description of ocean acidification and how AOA works.

Thanks

L50: 'Including through coral bleaching' a little clunky, maybe remove 'through'

We have removed this statement, it now states: While warming represents an imminent global threat which is already significantly impacting the natural environment (Hughes et al., 2017), ocean acidification poses an additional and equally significant threat to the marine environment.

L54: Could you say something about the changing Revelle Factor, and the potential for AOA to impact this?

We have now added the statement to the text: As CO2 is taken up by the ocean it changes its chemical equilibrium, reducing the carbonate ion concentration and decreasing pH, collectively known as ocean acidification. We have also added statement later in the introduction to say: Artificial Ocean Alkalization (AOA), through altering the chemistry of seawater, both enhances ocean carbon uptake (thereby reducing atmospheric CO2), while at the same time reversing ocean acidification and increasing the buffering capacity of the ocean.

L148-150: simply states "impact" which could be a bit vague. Could go in further and state that they will be investigating the impact on the "carbon cycle, global surface warming (2m surface air temperature), and response and ocean acidification response to the 4 different AOA experiments under the high (RCP8.5) and low (RCP2.6) emissions scenarios." Which is stated in lines 207-210. Regarding my comment above, it is worth exploring the potential experiment space, magnitude of alkalinity addition, lo cation, emission scenario, and the resulting impacts site specific/regional/global/ open ocean/coastal etc. What parts of this picture does your model/this paper deal with, what has already been done by others, and what is left to do? are other models needed?

The goals of this study of this focus on the global response to regional and seasonal AOA, therefore we don't explore are best ways to ameliorate local conditions through AOA. Regarding question of whether to explore experimental space further – the answer is yes there is a lot of work that needs to be done (please see the review by Renforth and Henderson (2017)) and these results are put into this context in the Results and Discussion section. AOA will also have addressed as part of the Carbon Dioxide Removal Model Intercomparison Project (CDR-MIP), which we are involved in.

We have now modified the paragraph to now say: In this work, we use a fully coupled ESM (CSIRO-Mk3L-COAL), which includes climate and carbon feedbacks, to investigate the impact of AOA on the carbon cycle, global surface warming (2m surface air temperature), and ocean acidification response to the global and regional AOA experiments under the high (RCP8.5) and low (RCP2.6) emissions scenarios

Methods:

Model seems appropriate for the scope of this paper. Clear description of the experimental design which seems appropriate to answer the research question proposed in the introduction. Could explain what the model outputs are? Also should mention the testing for seasonality? (mentioned in lines 538-554)

This model has been assessed in a number of studies already cited here and the outputs are consistent with standard Earth System Model outputs. As stated above the key outputs we consider are surface air and ocean temperature, and changes in the surface pH and aragonite saturation state. We also describe in the methods sections a number of other model prognostic variables (lines: 153-168) and provide references to individual model papers that describes components of the CSIRO Mk3L-COAL earth system model.

L162-164: Do you expect this assumption to hold up under elevated alkalinity? Could the rain ratio change?

Probably, this is already addressed in the lines 445-460, and studies suggest a small feedback.

L204: Fair assumption, but it is worth pointing out that alkalinity manipulation could be from carbonate dissolution or NaOH addition which would not induce and impact from iron and silicate. You are testing the fundamental impact intrinsic to all of these methods of C sequestration.

We agree, the sentence now reads: We do not consider the biogeochemical response to other minerals and elements that can be associated with the sourcing of alkalinity from the application of finely ground ultra-mafic rocks such as olivine and forsterite, nor dissolution processes required to increase alkalinity (e.g. Montserrat et al., 2017). Results and discussion:

L208 – 209: the sentence doesn't make sense, a typo somewhere? Yes, we have now

removed and response

L215: Why have you chose this addition rate? Also, should alkalinity not be expressed in equivalents rather than moles (and throughout)?

As stated, this value is very close to that used by Keller et al (2014) following Kohler (2014), who estimated a value of AOA based on globally shipping. While this allows a comparison of simulated values and a quasi-physical value, our work is more focussed on comparing and contrasting the responses to AOA for low and high emissions to regional and seasonal AOA. Regards units we have followed the convention used by Zeebe and Wolf-Gladrow (2005) and followed Keller et al (2014) and to ensure consistency with previous work.

L218-221: Fascinating, but why was the response different?

This is now addressed in the discussion

L232: "at" is missing

Addressed

L239: I think 'an overall' is missing before 525 ppm in the brackets

Added

L241: could you also give this as a % similar to how you did for RCP8.5

This is a good comment – however. This doesn't really make sense as the atmospheric value at end is less than at the beginning.

L251-254: This is really important...why was there an increase in export?... The 1% is an important outcome because it is the 'efficiency reduction' on the overall engineering system design.

Firstly, this is a really small number (<1%) and as stated occur in the Arabian Sea. In section 3.2.3 it now reads: The very small changes in export production in RCP2.6

were located in the Arabian Sea (not shown), likely driven by enhanced mixing in this region.

Could tables 1-3 be summarised in one table? I think it would make things a little clearer. This is a good and we have now combined these three into a single table

Line 331: could be explained more clearly rather than just "due to the Revelle factor" (see previous comment)

We have now rewritten this section to be clearer, it now states: In the 2020-2100 period, AOA under RCP2.6 led to much larger increases in surface pH and aragonite saturation state, more than 1.3 times, and more than 1.7 times that of RCP8.5 respectively (Table 4). These changes reflect the differences in the mean state associated with high and low emissions, specifically the difference between Alkalinity and Dissolved Inorganic Carbon (ALK-DIC), a proxy for ocean acidification (Lovenduski et al, 2015). As the values of DIC in the upper ocean are larger under RCP8.5 than RCP2.6, the difference between ALK and DIC (ALK-DIC) is smaller and the chemical buffering capacity of CO2 or Revelle Factor (Revelle and Suess, 1957) is less. This means that, for a given addition of ALK the increase in the upper ocean DIC will always be greater under RCP8.5 due to its reduced buffering capacity. Consequently, the changes in ALK-DIC with AOA are greater under RCP2.6 than RCP8.5, which translates to greater increases in pH and aragonite saturation state.

L374: 'This reflects the fact that' should be rewritten 'This is caused by the subduction processes..' or something similar.

We have now added through subduction to this statement

L396: 'Quite low', how low?

We have now removed Quite

L445-447: But could you speculate as you have in the previous sentence? How much would export have to change to make a material difference?

We would not like to speculate to do this as many of the processes are not well understood, instead we reference Matear and Lenton (2014) for a discussion of these processes and feedbacks.

L507: This doesn't quite ring with your abstract, which suggests that ocean acidification would be mitigated. Would it not partially ameliorate the impacts?

The last line of the abstract refers to low emissions, so they are consistent. Yes, it would ameliorate some of the impacts under RCP8.5, as shown, but its impacts would much less than under RCP2.6.

L599: 'Interestingly' is used a bit too often, it gets a bit jarring.

Reduced its usage

Figures:

All Figures are clear. Slightly too many for this type of manuscript. Could some be moved to the supporting information? Figures 11-12 are not referred to in the text.

We feel all figures are warranted, and we have added references to figures in the text.  

Please also note the supplement to this comment:
https://www.earth-syst-dynam-discuss.net/esd-2017-92/esd-2017-92-AC1-supplement.pdf

---

## Author Comment (AC2) · 24 Jan 2018

Response to Reviewer 2's comments for Assessing Carbon Dioxide Removal Through Global and Regional Ocean Alkalization under High and Low Emission Pathways by Lenton et al

Reviewer 2: Major comments:

Major comment 1: The changes in the land carbon uptake (table 2) in the AOA simulations based on the RCP2.6 are around 4 times higher than those of the simulations based on the RCP8.5. This is an important aspect because the variations in these

carbon fluxes determine the final state of the climate. That is why I think that these results should be discussed properly and the cause of this differential behaviour should be explained.

These differences are due to 2 main factors: (i) the temperature differences between RCP2.6 and RCP8.5. The mean SAT cooling over land under RCP2.6 is much larger (2x) this means that the decrease in carbon uptake would be larger RCP2.6 than RCP8.5; and (ii) as seen Zhang et al (2014a), the climate sensitivity of the land carbon-climate feedback under higher emissions is lower than other models' due to nutrient (N&P) limitation. This sensitivity in part explains why the response of the land carbon cycle is about half that reported in Keller et al (2014).

The text now states: . . .On the land, in the RCP8.5 simulation there was a smaller reduction in carbon uptake than in RCP2.6 (Table 1), due to larger decreases in surface air temperature (SAT) over land in RCP2.6 than RCP8.5 (∼2x; see Section 3.1.2). The land carbon cycle response was also smaller under high than low emissions due to nutrient limitation being reached, thereby limiting the effect of $CO_2$ fertilization (Zhang et al, 2014a).

Major comment 2: The statements given between the line 285 and 289 are really confusing. On the one hand, it reads as the temperature change in the RCP8.5 experiment is higher than the one associated with the RCP2.6, which is not what I see in the numbers. And on the other hand, making reference to "potentially reflecting feedbacks" in order to explain this cooling signal does not help to understand the signal. Instead, it confuses the reader. Please explain properly how these feedbacks affect the results.

This was a mistake and has now been corrected. We have now clarified the text and removed the reference to feedbacks which was not correct, please see the comment below showing that the disparate responses are primarily due to differences in atmospheric $CO_2$ growth rate, please the response to Major Comment 3 (next) for more detail.

Major comment 3: The reduction in atmospheric CO2 concentrations by 2100 associated with the AOA scenarios under RCP8.5 emissions (app. 84 ppm) is higher than the one associated with the AOA scenarios conducted under the RCP2.6 (app. 40 ppm). Yet, the mitigated warming in the AOA simulations under RCP2.6 is higher than those conducted under the RCP8.5. This is one of the main findings of this publication, however, there is not any discussion/explanation of this result. Only stating what the model delivers is not enough, since it could be a model artefact, the signal might not be caused by AOA, etc. The RCP8.5 and 2.6 scenarios have atmospheres with quite different levels of CO2, which might lead to differences in the CO2-forcing response to changes in CO2 levels. Not only that, but also the RCP8.5 and 2.6 scenarios differ in the assumed land use and the sea ice extent by the end of this century. This might also cause changes in albedo and therefore in the cooling response due to changes in forcing.

There are a number of mechanisms that may explain the differential response of the cooling which is larger under RCP2.6 than RCP8.5, these including the CO2 vs outgoing long wave radiation (OLR) log relationship, and land and ocean albedo changes. We find that while these may play a minor role, the major driver of these differences are due to differences in atmospheric CO2 growth rate between RCP2.6 and RCP8.5.

We have now added the following statement to the text: . . .In the period 2081-2100 we see larger mean changes in SAT under RCP2.6 than RCP8.5 primarily due to differences in atmospheric CO2 growth rate. Krasting et al. (2014) showed that the slower rate of emissions, the lower the radiative forcing response. This occurs in response to the timescales associated with the uptake of heat and carbon. Consequently, under RCP8.5 the atmospheric CO2 growth rate is much faster than RCP2.6, leading to a strong radiative forcing response. This explains why, despite a larger reduction in atmospheric CO2 concentration under RCP8.5, the biggest reduction in global mean SAT occur under RCP2.6. . .

Ref: Krasting, J. P., Dunne, J. P., Shevliakova, E., and Stouffer, R. J.: Trajectory sensitivity of the transient climate response to cumulative carbon emissions, Geophys Res Lett, 41, 2520-2527, 10.1002/2013gl059141, 2014.

Major comment 4: Between the lines 325 and 334 an explanation to the differential pH and aragonite saturation state responses between simulations is given. This explanation seems confusing and it refers to the other main finding of this publication. Because of this I think that it requires some supporting figures (which could be added into the supplementary information) and some extra work in order to clarify the message. I suggest to look at the buffer factors and the effects of AOA under the two different DIC/ALK regimes associated with the RCP8.5 and 2.6 scenarios. More information can be found in the paper by Egleston et.al. (2010) (http://onlinelibrary.wiley.com/doi/10.1029/2008GB003407/abstract). We have rewritten this paragraph to better capture our message make it more accessible.

It now states: In the 2020-2100 period, AOA under RCP2.6 led to much larger increases in surface pH and aragonite saturation state, more than 1.3 times, and more than 1.7 times that of RCP8.5 respectively (Table 4). These changes reflect the differences in the mean state associated with high and low emissions, specifically the difference between Alkalinity and Dissolved Inorganic Carbon (ALK-DIC), a proxy for ocean acidification (Lovenduski et al, 2015). As the values of DIC in the upper ocean are larger under RCP8.5 than RCP2.6, the difference between ALK and DIC (ALK-DIC) is smaller and the chemical buffering capacity of CO2 or Revelle Factor (Revelle and Suess, 1957) is less. This means that, for a given addition of ALK the increase in the upper ocean DIC will always be greater under RCP8.5 due to its reduced buffering capacity. Consequently, the changes in ALK-DIC with AOA are greater under RCP2.6 than RCP8.5, which translates to greater increases in pH and aragonite saturation state.

Minor comments: L16, L27 and L561: "is capable of" gives the impression that AOA has not real big limitations to be implemented which is not the case, please modify the wording This wording is correct, AOA is capable and is analogous to alkalinity addition

that occurs over geological timescales and this has been hypothesised to play a role in glacial-interglacial timescales.

L18, L19: there are acronyms which the reader might have never seen in the abstract, please spell them out or remove

We have now spelt out CO2 and RCP

L25: "lower" and "higher" emissions than what? I think that you meant "low" and "high"

This is correct, we wanted to be more generic than just RCP 2.6 and 8.5, particularly as Shared Socioeconomic Pathways (SSPs) will be used CMIP6.

L26: our simulations show that AOA during the period ... ; in any case I do not think that this very last sentence in the abstract is needed

With respect, we think that this is an important statement to make.

L46: ... could help to ...

Corrected

L53-54: CO2 that enters the ocean does not react with seawater to reduce the carbonate ion concentration, please reconsider this statement and use correct grammar

This has now been changed to say: As CO2 is taken up by the ocean it changes its chemical equilibrium, reducing the carbonate ion concentration and decreasing pH, collectively known as ocean acidification.

L59: ...changes in calcification...

Corrected

L60: are you sure that ocean acidification alters nutrient availability

Yes e.g. Shi et al (2010) Shi et al (2010) Effect of Ocean Acidification on Iron Availability to Marine Phytoplankton, Science 327, 676 (2010);

L62: please change order of publications

They are already in name order

L69: semicolon needed?

Removed

L77: weathering of minerals play a crucial role in modulating the state of the climate in geological timescales, please write an assertive statement

Removed may

L91: reviewed

This should be present tense, I believe

L92-95: way too long sentence, please simplify and split it

This has now been split

L98: Did Kohler used one or several models? Changed to singular

L110: ocean only without the hyphen

Corrected

L110: and they showed

Corrected

L111: high CO2 emission

Corrected

L114: also concluded that

This should be present tense, I believe

L115, L118, L131, ...: impacts of

Changed

L124: from a high

Changed

L126: it would be required

Changed

L127: and it would come

Changed

L134: 78 ppm between brackets might look better

Changed

L134: a net atmospheric cooling

Changed to surface air temperature

L139: "to be very large" - (very) large in what respect? please clarify

The sentence now reads: Capturing these feedbacks is critical as they have the potential to significantly increase atmospheric CO2 concentrations (Jones et al., 2016).

L141: currently assume (instead of "utilize")

Changed

L141 to L145: way too long sentence, please simplify and split it

Changed to: Furthermore, the feasibility of these approaches which are increasingly questioned due in part to limited land (Smith et al., 2016), whereas the potential CDR capacity of the oceans is orders of magnitude greater (Scott et al., 2015).

L149: and surface warming L149: questions L147 to L152: I think that the novelty of this study could be better emphasise. In any case, this last paragraph is crucial and

therefore it should be improved since it does not read well.

The above three comments have been addressed in response to Reviewer 1 and the paragraph now reads: In this work, we use a fully coupled ESM (CSIRO-Mk3L-COAL), which includes climate and carbon feedbacks, to investigate the impact of AOA on the carbon cycle, global surface warming (2m surface air temperature), and ocean acidification response to the global and regional AOA experiments under the high (RCP8.5) and low (RCP2.6) emissions scenarios.

L158: extra dot after citations?

Changed

L160, L163, ...: please remove the brackets in those citations which are subjects of the sentences, this occurs several times in this manuscript

Changed

L164 to L166: does this sentences really add any relevant information? Such a feature of the model is basic to conduct this study

Yes, we agree but it does provide confidence in the tool we are employing in this study – as seen by Review 1's comments

L171 to L172: the land carbon cycle currently has too many uncertainties to state something in such an assertive manner, please consider to modify this or even remove it

We have removed realistically

L185: from 2006 onwards, ...

Corrected

L186: corresponding to the Representative ...

Corrected

L218: Subpolar addition

Corrected

L232: is seen in 2100

Corrected

L233 to L235: if you mention this feature of the modelling tool, please explain the associated consequences for the simulations of AOA

We have discussed these results in the manuscript and identify why the sensitivity of the land carbon uptake, particularly under high emissions is less than other studies.

L242 and L256: "more than compensates" and "more than offset" are really confusing ways of describing the obtained values, please clarify

We have removed more than in both of these instances

L247: 50% instead of 1.5 maybe?

Corrected

L253: total ocean uptake ...

Corrected

L254 to L258: this explanation reads really confusing, please clarify

This has now been changed to say: The simulated cooling drove both a reduced net primary production, leading to reduced carbon uptake, and an increase in carbon retention associated with a reduction in heterotrophic respiration. However, overall, the net decrease in land carbon uptake means that in the response to AOA globally the reduced net primary production dominated.

L266: addition studies such as Ilyina ... which demonstrated ...

Corrected

L270: 181 PgC is in ... (instead of was)

Corrected

L277: I think the authors meant "positive denotes enhanced uptake" (instead of "negative")

Yes – this is corrected thank you

L288: "large" twice in the sentence

Corrected

L294: "projected" instead of "anticipated"

Corrected

L295: why is this publication here cited?

Removed

L298: standard deviations with respect to what? what is this (1 - sigma)? Please clarify

The caption has now been improved, and now reads: Table 1 The differences in global mean surface air temperature in the period 2081-2100 (2090) and associated standard deviation (1-) (K; SAT; 2m) for the four different AOA experiments for each emission scenario, relative to the same emission scenario with no AOA.

L302 to L304: please consider to reformulate these sentences since "variability" might refer to many different things (e.g. inter annual, inter model, model internal, ...). In any case I think that "variability" is not really the term to use since what is described here are differences between simulations.

The sentence now reads: Within each of the scenarios, there are some differences in the magnitude of the cooling within the four different AOA experiments; however, these are smaller than the interannual variability over the last two decades of the simulations.

L308: mean surface cooling

Corrected

L318: What is the point of this statement and citation? The pH and aragonite saturation state correlate really well as I can see in the figures.

Yes but the impacts are different and this motivates why we are interested looking at both aragonite saturation state and pH.

L321: despite the return

Corrected

L344 and L377: the citation here to Groeskamp et.al. seems unfounded

Removed

L348 and L349: Please elaborate on this so that the reader understand the context, e.g. discuss how this change in pH might (or not) matter, ...

We have modified this sentence to now say: To put these changes into context, the estimated decrease in pH since the preindustrial period is 0.1 units (Raven et al., 2005), and is responsible for already detectable changes in the marine environment (Albright et al., 2016).

L380: How can one of the experiments (AOA_ST) reflect the timescales of the circulation of the subtropical gyres? Please explain this.

This sentence now reads: In the case of AOA_ST, this reflects the timescales associated with the longer residence time of upper ocean waters in the subtropical gyres.

L382: ice covered (instead of "non-ice-free")

Changed

L386: by 2100 (instead of "in")

Corrected

L387 and L388: please clarify this, is not understandable

This is now clarified to say: Specifically, for AOA_G we see 31% remains in the upper ocean and for AOA_T and AOA_ST: 34%, while for AOA_SP: 22-24% remains in the upper ocean which (as anticipated) is lower than in other regions.

L404: ...seen in the...

Corrected

L421 to L425: please work on the grammar of these sentences

Corrected

L442: ... in the ratio ...

Corrected

L444: Dot missing

Added

L445: ... remain poorly...

Corrected

L457 to L459: why do you obtain this result?

We have added: likely driven by enhanced mixing in this region.

L463: remove (SAT)

Removed

L468 to L470: why do you obtain this result?

This very much reflects the period over which the mean changes were calculated, and

the simulated large variability in SAT in this region, which now stated in the text.

L538 to L554: why no figures are shown in this section on seasonality to support this discussion? Also, only AOA is implemented in the summer season under RCP8.5 emissions, which does not seem to me enough to explore the effects of seasonality.

This is a little confusing as the Reviewer wishes us to remove figures and now requests more. We don't present results here, as it is a sensitivity experiment, rather than a major result.

L560: please remove (COP21)

Removed

L593 to L595: What do you mean? Please clarify this.

It now states: However, AOA under the RCP2.6 emissions scenario changes the roles played by the ocean and land in carbon uptake as compared with the scenario of RCP2.6 with no AOA, resulting in a reduced uptake in the terrestrial biosphere and increased uptake in the ocean.

L605: double "that"

Corrected

L620: mention "preindustrial period" and remove (1850) reads better

Changed

L621: cases (subject?) leads ...

Corrected

L633: for the role

Corrected

L638: ...therefore it needs ...

Added

L642 and L645: Earth system (instead of earth system)

Corrected L649: please put "e.g. mesocosm experiments" between brackets

Added

Please keep an eye on the format in which the references are given and be consistent with it.

Done

 

Please also note the supplement to this comment:
https://www.earth-syst-dynam-discuss.net/esd-2017-92/esd-2017-92-AC2-supplement.pdf

---

## Author Comment (AC3) · 24 Jan 2018

Response to Reviewer 3's comments for Assessing Carbon Dioxide Removal Through Global and Regional Ocean Alkalization under High and Low Emission Pathways by Lenton et al

Comments Reviewer 3

The Lenton et al., study investigates the impacts from adding artificial alkalinity to the oceans using the model CSIRO under 2 different emission pathways - RCP2.6 and RCP8.5. It was a really well done and interesting study to read technically, however

my main comment is that the way the paper is currently structured makes it confusing to read. For example, each paragraph jumps back and forth between RCP26 and RCP85 making the story line hard to follow. I suggest setting up the story for one of the emissions pathways and then comparing to that one for the other pathway. It would also be useful to set up the chemistry in a little more detail or reiterate the paragraph in the intro. This would be useful when explaining why adding alk under a 2.6 scenario is more effective.

At this stage, we do not feel that major rewrite or reordering of the paper is warranted. It is clear from other studies is that AOA will reduce OA and global warming; what is more interesting is whether the response to the same amount of AOA differs between emissions scenarios. This is the main focus of the study hence it does not make sense to restructure the paper as suggested. Instead we have gone through the paper to ensure that it is clearer and easier to follow.

It would also be useful to set up the chemistry in a little more detail or reiterate the paragraph in the intro. This would be useful when explaining why adding alk under a 2.6 scenario is more effective.

Please see the response to Reviewer 2.

Lastly, section 3.1 was confusing (you may want to expand on the methods section to make this section clearer). For each run you added 0.25Pmol/yr of alkalinity but then I read in ln216 that the magnitude of the increase in alkalinity is dependent on where it was added. Is the 0.25Pmol/yr added to all the boxes? or is it divided up between the boxes for a total of 0.25Pmol/yr? Can you put everything in the same units to be constant?

We apologise for any ambiguity and have now clarified this section, it now states: For each emissions scenario, we simulated four different AOA experiments, which all had the same 0.25 Pmol/yr of alkalinity added. In the case of the regional experiments the per surface values were larger than the case of global addition.

Minor comments: ln50: "including through coral bleaching" - not clear what this means

We have now removed this statement.

ln79-80: This sentence seems out of place.

We have now removed this sentence.

ln149: what do you mean by impact?

We have now been more explicit and the sentence now says: In this work, we use a fully coupled ESM (CSIRO-Mk3L-COAL), which includes climate and carbon feedbacks, to investigate the impact of AOA on the carbon cycle, global surface warming (2m surface air temperature), and ocean acidification response to the global and regional AOA experiments under the high (RCP8.5) and low (RCP2.6) emissions scenarios.

ln158: extra period between feedbacks and references

Corrected

ln230: the first sentence does not make sense.

Rewritten it now states: The large atmospheric CO2 concentration at 2100 under RCP8.5 reflects the large projected increase in emissions during this century, while under RCP2.6 a similar atmospheric concentration of CO2 is seen in 2100 as at the beginning of the simulation (2020) (Figure 2a).

ln250: why is there a difference in export?

The text now reads: ...Consistent with Keller et al. (2014) and Hauck et al. (2016) the simulated changes in ocean export production were very small ($\sim$0.2 PgC) under RCP8.5 and due to small changes in ocean state, e.g. stratification. Under RCP2.6, it was slightly larger at 1.2 PgC, but still less than 1% percent of the total ocean uptake increase simulated under AOA, due to small changes in ocean state in a more stratified ocean...

Section 3.1.2: I don't understand how soil-moisture feedbacks are influencing temperature in this case. Temperature is more variable on land than over the ocean, could it be simply that?

We apologise for the confusion we have removed this section and attribute these changes to the differences different in atmospheric CO2 growth rate.

The section now states: ...In the period 2081-2100 we see larger mean changes in SAT under RCP2.6 than RCP8.5 primarily due to differences in atmospheric CO2 growth rate. Krasting et al. (2014) showed that the slower rate of emissions, the lower the radiative forcing response. This occurs in response to the timescales associated with the uptake of heat and carbon. Consequently, under RCP8.5 the atmospheric CO2 growth rate is much faster than RCP2.6, leading to a strong radiative forcing response. This explains why, despite a larger reduction in atmospheric CO2 concentration under RCP8.5, the biggest reduction in global mean SAT occur under RCP2.6...

Ref: Krasting, J. P., Dunne, J. P., Shevliakova, E., and Stouffer, R. J.: Trajectory sensitivity of the transient climate response to cumulative carbon emissions, Geophys Res Lett, 41, 2520-2527, 10.1002/2013gl059141, 2014.

Table4: relative to what?

It now reads: Table 2 The differences in surface value of aragonite saturation state and pH between the AOA experiments for each emission scenarios in 2100 relative to the emissions scenario with no AOA.

Figures: relative to what?

We have added text to each of the captions to clarify

Figures: There are a lot of figures which I'm not sure add much to main text. You could simply the story in the text and only look at a few and toss the rest into the supplement. Or could you combine the 2.6 and 8.5 figures into 1? Could you do a difference between them? That would visually show the reader where the differences

between scenarios lie....

While this seems attractive, we think that there is value in keeping these figures. Furthermore, we do not see a simply way of combining these into 8 panel figures, nor does doing the differences make much sense, as differences of differences is quite confusing.

Section 3.2.5: I think the figures referenced are not the correct figures. Plus I think a description of figures 11 and 12 are missing.

Thank you for this – we have now ensured that the figures are referenced correctly, and switched the order to better reflect the order they are appear in the text. I see that the previous reviewers picked up a bunch of typos

Corrected.

Please also note the supplement to this comment:
https://www.earth-syst-dynam-discuss.net/esd-2017-92/esd-2017-92-AC3-supplement.pdf